# Synthetic lethality between HER2 and transaldolase in intrinsically resistant HER2-positive breast cancers

Yi Ding[1], Chang Gong[2], De Huang[1], Rui Chen[1], Pinpin Sui[3], Kevin H. Lin[1], Gehao Liang[2], Lifeng Yuan[1], Handan Xiang[1], Junying Chen[2], Tao Yin[1], Peter B. Alexander[1], Qian-Fei Wang[3], Er-Wei Song[2], Qi-Jing Li[4], Kris C. Wood[1] & Xiao-Fan Wang[1]

Intrinsic resistance to anti-HER2 therapy in breast cancer remains an obstacle in the clinic, limiting its efficacy. However, the biological basis for intrinsic resistance is poorly understood. Here we performed a CRISPR/Cas9-mediated loss-of-function genetic profiling and identified *TALDO1*, which encodes the rate-limiting transaldolase (TA) enzyme in the non-oxidative pentose phosphate pathway, as essential for cellular survival following pharmacological HER2 blockade. Suppression of TA increases cell susceptibility to HER2 inhibition in two intrinsically resistant breast cancer cell lines with HER2 amplification. Mechanistically, TA depletion combined with HER2 inhibition significantly reduces cellular NADPH levels, resulting in excessive ROS production and deficient lipid and nucleotide synthesis. Importantly, higher TA expression correlates with poor response to HER2 inhibition in a breast cancer patient cohort. Together, these results pinpoint TA as a novel metabolic enzyme possessing synthetic lethality with HER2 inhibition that can potentially be exploited as a biomarker or target for combination therapy.

[1] Department of Pharmacology and Cancer Biology, Duke University Medical Center, Durham, NC 27705, USA. [2] Breast Tumor Center, Sun Yat-Sen Memorial Hospital, Sun Yat-Sen University, Guangzhou 510120, China. [3] Key Laboratory of Genomic and Precision Medicine, Collaborative Innovation Center of Genetics and Development, Beijing Institute of Genomics, Chinese Academy of Sciences, Beijing 100101, China. [4] Department of Immunology, Duke University Medical Center, Durham, NC 27705, USA. Correspondence and requests for materials should be addressed to K.C.W. (email: kris.wood@duke.edu) or to X.-F.W. (email: xiao.fan.wang@duke.edu)

The human epidermal growth factor receptor 2, or HER2, was discovered in 1985[1–3]. Overexpression of HER2 occurs in 25–30% of breast cancer patients and is associated with aggressive disease[4,5]. Therapies targeting HER2, including monoclonal antibodies (trastuzumab and pertuzumab), a small molecule kinase inhibitor (lapatinib) and an antibody-drug conjugate (trastuzumab emtansine), have significantly prolonged the overall survival of HER2-positive breast cancer patients in both the adjuvant and metastatic settings[6–9]. Despite these major advances, up to 30% of patients with high risk early stage HER2-positive breast cancer will develop recurrence following treatment with 1 year of HER2 targeted therapy, suggesting intrinsic resistance[10]. Approximately 20% of patients with HER2-positive metastatic breast cancer fail to respond initially to the most advanced HER2 targeted therapies[11], and almost all patients who initially respond relapse within 2 years[12–14], suggesting the presence of both intrinsic and acquired resistance to targeted therapy.

Previous studies on anti-HER2 resistance have uncovered three general mechanisms by which cancer cells escape HER2 inhibition. First, the expression of certain surface proteins, such as MUC4 and p95HER2, can directly interfere with antibody binding to HER2[15,16]. Alternatively, mutation of downstream signal transducers can function to sustain proliferative and pro-survival signaling pathways, including loss of the PTEN tumor suppressor and constitutive activation of PI3K[17–19]. Third, tumor cells can increase expression of alternative RTKs such as IGF1R in order to bypass HER2 inhibition and maintain intracellular signaling flux[20–22]. Although these studies provide insight into anti-HER2 resistance mechanisms and possible targets to be exploited pharmacologically, many were performed using acquired resistance models derived by chronically treating cells with increasing doses of HER2 inhibitors. Therefore, those models may not be relevant to the 20% of metastatic HER2-positive breast cancer patients who exhibit intrinsic resistance, failing to respond to anti-HER2 therapy from the beginning[12].

The pentose phosphate pathway (PPP) is a primary biosynthetic pathway comprised of two sequential series of reactions: two oxidative steps, which irreversibly generate ribose-5-phosphate (R5P) for nucleic acid synthesis and NADPH for reductive anabolic pathways and cellular redox balance; and the non-oxidative portion, which reversibly converts pentose phosphate into glycolytic intermediates that can then fuel glycolysis or gluconeogenesis. The reversible nature of the non-oxidative PPP enzymes enables cells to generate the appropriate ratios of nucleic acids and NADPH for proliferation and survival[23]. In fast dividing cancer cells, enzymes mediating both phases of the PPP are expressed at high levels to support nucleotide synthesis and balance increased oxidative stress[24–29].

In order to discover mechanisms mediating intrinsic anti-HER2 resistance and identify novel therapeutic targets, we performed functional CRISPR/Cas9 genetic profiling in MDA-MB-361 cells, which are intrinsically resistant to HER2 inhibition. From this screen, we identified the metabolic enzyme transaldolase (TA), which catalyzes the rate-limiting step in the non-oxidative PPP, as essential for cells to survive under lapatinib treatment. Isotopic label-based metabolic profiling indicates that HER2 inhibition systematically alters cellular metabolism, such that metabolic flux through the non-oxidative PPP becomes essential for cancer cells to survive lapatinib treatment. Because TA expression effectively stratifies the outcome of HER2-targeted therapy in breast cancer patients, TA may represent a novel biomarker and new target for the treatment of recalcitrant HER2-positive breast cancer.

## Results

**Transaldolase was identified from a functional genetic screen.** Loss-of-function genetic screening using the CRISPR/Cas9 system has been proven to be a powerful method for identifying key molecules mediating complex biological processes such as tumor progression[30–32]. We exploited this approach to identify genes whose loss-of-function can sensitize cells that are intrinsically resistant to HER2 inhibition. Specifically, we used a single-guide RNA (sgRNA) library targeting 378 target genes (5 guides per gene) that covers most major growth signaling and metabolic pathways, with 50 non-targeting controls, as reported previously[33]. In order to focus on the clinical problem of intrinsic drug resistance, we used MDA-MB-361 cells, which are derived from a HER2-amplified metastatic breast cancer patient and insensitive to lapatinib[34]. As in previously published CRISPR/Cas9 screening procedures[30], cells were infected and selected with puromycin for 7 days, at which point ($T = 0$) DNA samples were collected to measure genomic sgRNA integration. Then, pools of cells carrying sgRNA constructs were split equally into four biological replicates and further treated with either vehicle (DMSO) or 1 μM lapatinib for 2 weeks (Fig. 1a). The lapatinib concentration used for this experiment is sufficient to induce 90% growth inhibition in lapatinib-sensitive (BT-474) cells, but only 20% inhibition in MDA-MB-361 cells (Fig. 1b).

In this experimental setting, cells carrying sgRNAs targeting genes that are essential for viability would be depleted under both the vehicle and lapatinib treatment conditions (essential genes). In contrast, sgRNAs depleted to a significantly greater extent in lapatinib-treated cells compared to vehicle might be expected to target genes that exhibit synthetic lethality with HER2 inhibition (sensitizer genes). We applied a previously defined algorithm to weigh each set of five guides targeting the same gene and rank all genes[33]. Depletion of essential genes was consistent between the two biological duplicates (Supplementary Fig. 1a), suggesting reliable sample processing and data analysis. As expected, we observed a more significant depletion of genes in essentiality analysis compared to sensitivity analysis[33] (Supplementary Fig. 1b–c).

Based on the analysis described above, we calculated and plotted the average fold change of two biological replicates treated with DMSO versus lapatinib. While most of the genes were depleted to similar extents under both conditions, two genes—IGF1R and TALDO1—were much more significantly depleted after lapatinib treatment (Fig. 1c). By using [lapatnib-2-week] to [DMSO-2-week] ratios to rank sensitizer genes, the five highest ranked sensitivity genes were IGF1R, TALDO1, GATA3, TBX3, and PTK2 (Fig. 1c). IGF1R has been reported in multiple studies as a key mediator of bypass-resistance to HER2 inhibition;[20–22] the top ranking of IGF1R thus validates the capacity of our screening approach to uncover bona fide regulators of tumor cell responsiveness to lapatinib. Whereas PTK2 (also called FAK) was also reported to sensitize cells to lapatinib[35], TALDO1, GATA3, and TBX3 have not been previously reported to function in this manner. TALDO1 encodes the transaldolase (TA) metabolic enzyme which catalyzes a key non-oxidative reaction in the PPP (sedoheptulose-7-phosphate + glyceraldehyde 3-phosphate ⇌ erythrose 4-phosphate + fructose 6-phosphate). GATA3 and TBX3 encode transcription factors that are frequently mutated and highly expressed in breast cancer, respectively[36,37]. To investigate these further, we plotted depletion percentages of the top five candidates in samples treated with either DMSO or lapatinib for 2 weeks. While sgIGF1R targeting constructs were consistently depleted to the greatest extent (Supplementary Fig. 1d), the other sgRNAs were also depleted in lapatinib-treated

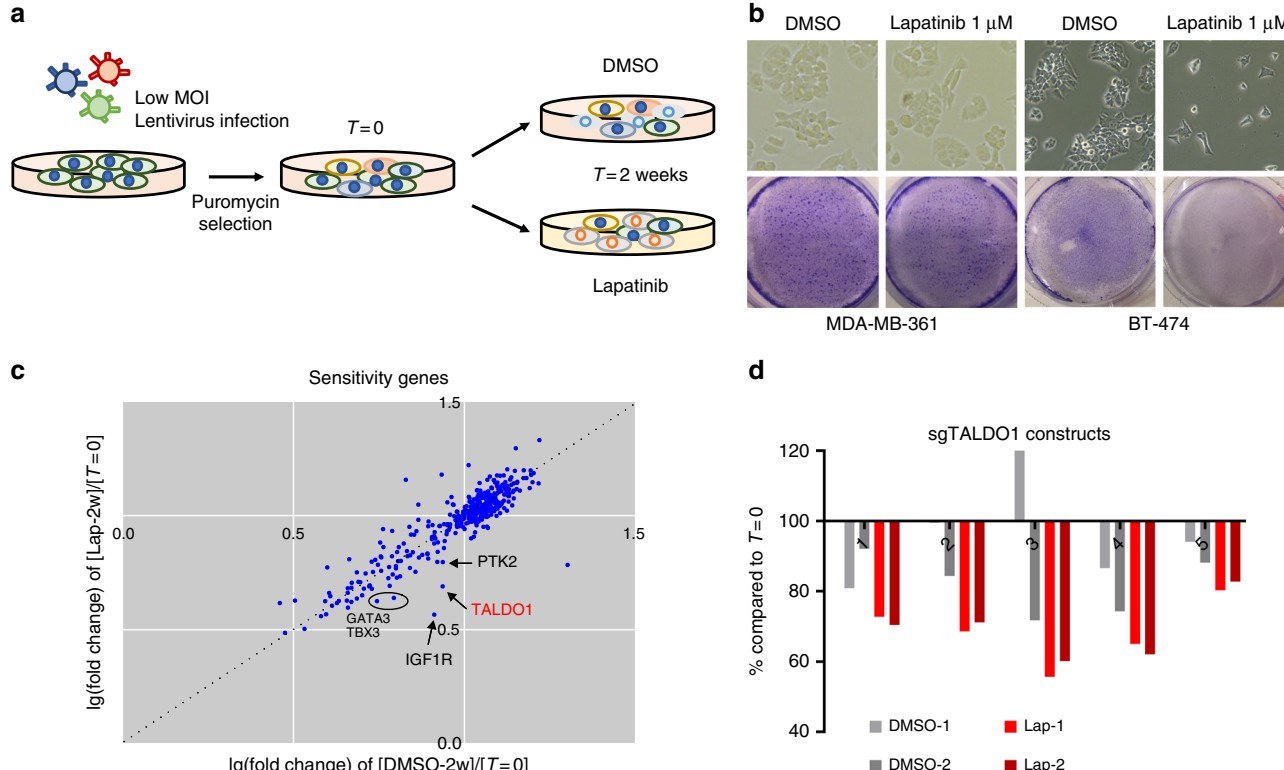

**Fig. 1** Pooled CRISRP/Cas9 screening strategy identifies sensitivity genes. **a** Schematic representation of pooled CRISPR/Cas9 screening completed in MDA-MB-361 cells. Samples were collected at the indicated time points. **b** Bright-field micrographs and crystal violet staining of cultured MDA-MB-361 and BT-474 cells treated with DMSO or 1 μM lapatinib. **c** Strategy to identify sensitivity genes. The average of two biological repeats each gene depletion: x-axis as DMSO (2 weeks) compared to $T = 0$, y-axis as lapatinib (2 weeks) compared to $T = 0$, were plotted. *IGF1R*, *TALDO1*, *GATA3*, *TBX3*, and *PTK2* are indicated as the most differentially depleted genes after lapatinib treatment. **d** Depletion percentages of individual sgRNA constructs in indicated samples compared to $T = 0$

cells with some variance between biological replicates and targeting constructs (Fig. 1d and Supplementary Fig. 1e–g). Together, these results suggest that our CRISPR/Cas9 genetic profiling data corroborate previously known molecules involved in anticancer drug resistance and identify new candidates for further investigation.

**TA deficiency confers sensitivity to lapatinib.** In order to validate our genetic profiling results, we next cloned individual sgRNA constructs targeting the *TALDO1*, *GATA3*, and *TBX3* genes, as well as *IGF1R* to be used as a positive control. As expected, *IGF1R* deficiency was able to significantly increase breast cancer cell sensitivity to lapatinib in a three-day dose-response assay; however, among the newly identified genes, only *TALDO1* depletion conferred similar sensitivity in this assay (Supplementary Fig. 2a–d). For this reason, we narrowed our focus to the *TALDO1* gene product TA. To further verify these CRISPR/Cas9 results and assess TA's function in enhancing vulnerability to HER2 inhibition, we utilized shRNA-mediated TA knockdown as a second independent loss-of-function methodology. First, we verified that TA deficiency increases lapatinib sensitivity (Fig. 2a). Then, we analyzed MDA-MB-361 cell numbers following a four-day treatment with either DMSO or 4 μM lapatinib, which is the concentration with the greatest difference between sgNT and sgTA based on the dose-response curve (Fig. 2a and Supplementary Fig. 2b). Whereas DMSO-treated cells lacking TA can still proliferate, TA knockdown combined with HER2 inhibition results in a complete absence of cell growth (Fig. 2b). Similar results were obtained using a second independent HER2-positive, lapatinib-resistant cell line MDA-MB-

453. This cell line has a similar IC50 to lapatinib (Supplementary Fig. 2e). Treating cells with 1 μM lapatinib significantly reduced the survival of TA-deficient cells (Fig. 2c). Together, these results confirm that TA functions to maintain cell growth after HER2 blockade in breast cancer cell lines that are intrinsically unresponsive to HER2 inhibition.

Next, we explored whether TA deficiency might further sensitize lapatinib-sensitive cell lines, or alternatively might overcome resistance in acquired resistance models. For this, we generated stable cell lines expressing shNT or shTA constructs in the lapatinib-sensitive HER2-positive cell lines BT474, AU565, and UACC812, as well as in the acquired resistance rBT474 and rAU565 cell models developed by chronically treating either BT474[38] or AU565 parental lines with lapatinib. However, TA deficiency did not significantly sensitize any of these cells to HER2 inhibition (Fig. 2d and Supplementary Fig. 2f–h). Based on these results, we conclude that the ability of TA suppression to promote lapatinib sensitivity is likely specific for intrinsically resistant cells.

**TA expression reflects patient responsiveness to anti-HER2 therapy.** After determining that TA depletion overcomes intrinsic lapatinib resistance in breast cancer cell lines, we next asked whether basal TA expression correlates with intrinsic sensitivity. To test this, we measured TA mRNA and protein expression using a panel of HER2-positive breast cancer cell lines. TA mRNA levels were significantly higher in the two intrinsically resistant cell lines (MDA-MB-361 and MDA-MB-453) compared to their lapatinib-sensitive counterparts (UACC812, BT474 and

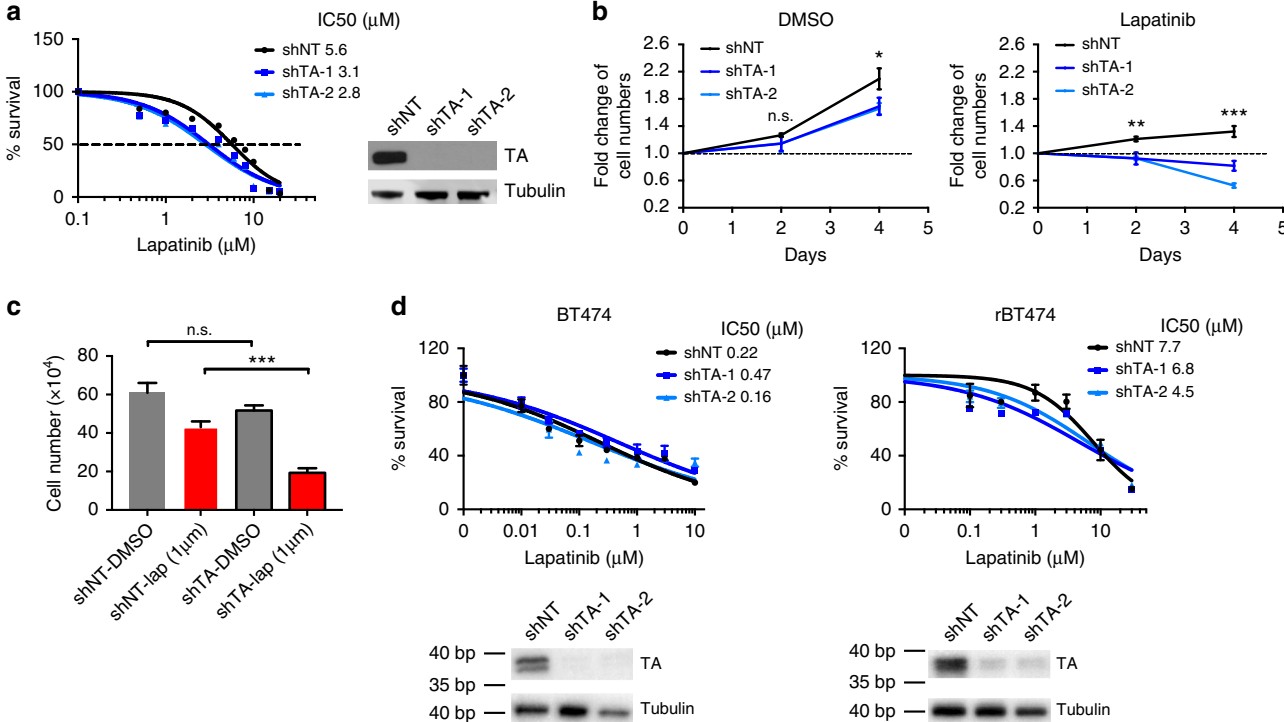

**Fig. 2** TA mediates sensitivity to lapatinib in intrinsically resistant breast cancer cells. **a** Dose–response curves of MDA-MB-361 cells carrying shNT or shTA constructs. ($n = 4$). Western blot showing shRNA knockdown efficiency. **b** Fold change of cell numbers at the indicated time points in shNT or shTA cells treated with DMSO or 4 µM lapatinib ($n = 3$). **c** Numbers of MDA-MB-453 cells cultured with DMSO or 1 µM lapatinib for 3 days and co-treated with either shNT or shTA ($n = 3$). **d** Dose-response curves of BT474 and rBT474 cells carrying shNT or shTA constructs ($n = 4$). Western blot showing shRNA knockdown efficiency. IC50s were calculated using Prism curve fit (variable slope) default program. Error bars denote the SD; n.s., not significant; *$P < 0.05$; **$P < 0.005$; ***$P < 0.001$ by unpaired, two-tail, student $t$-test

HCC1954; Fig. 3a). TA protein expression was proportional to mRNA expression except in MDA-MB-361 cells, which expressed low levels of TA in both the absence and presence of lapatinib (Fig. 3b). TA is a known component of the PPP, which is activated to support the metabolic needs of various types of cancer cells[23]. Interestingly, the expression of several other PPP enzymes increased with lapatinib treatment in sensitive BT-474 cells, but not in resistant MDA-MB-361 cells (Supplementary Fig. 3a). Together, these results suggest that TA-mediated metabolic flux through the PPP may play an important role in the breast cancer cellular response to HER2 inhibition.

To more directly assess the clinical relevance of these findings, we analyzed TA expression in breast tumors using both publicly available online databases and tissue samples harvested from tumor biopsies. We first used an established online database derived from GEO microarray data[39] to investigate the possible correlation between TA and relapse-free survival (RFS). Among HER2-positive patient samples ($n = 252$), this analysis revealed that higher TA expression is associated with poor patient outcome (Fig. 3c). To directly interrogate this correlation at the protein level, we performed immunohistochemical staining using 44 HER2-positive patient tumor biopsies that were harvested both before and after 4–8 cycles of neoadjuvant trastuzumab therapy. Samples were designated as either resistant or sensitive based on patient response to treatment (described in experimental procedures). Importantly, resistant tumors expressed significantly higher levels of TA compared to tumor samples obtained from patients who responded to trastuzumab therapy (Fig. 3d). TA expression did not increase after several cycles of treatment in either responders or non-responders (Supplementary Fig. 3b), indicating that while its initial level is predictive of response to anti-HER2 targeted therapy, it is unlikely to play a

general role in acquired resistance, consistent with our findings in cellular models of acquired resistance. Because TA expression prior to treatment is reflective of a patient's eventual response to HER2 inhibition, this metric may prove useful as a biomarker that can be used to assess a patient's suitability for receiving anti-HER2 targeted therapy.

**Non-oxidative PPP is essential for viability in the presence of HER2 inhibition.** Transketolase (TK) catalyzes two important reactions immediately adjacent to TA in the non-oxidative PPP (Supplementary Fig. 4a). To further investigate the necessity of PPP metabolism for breast cancer cell viability, we used the antimetabolite oxythiamine (OT) to pharmacologically inhibit TK. OT treatment synergized with HER2 inhibition at a level comparable to shRNA-mediated TA knockdown (Supplementary Fig. 4b), validating the importance of PPP flux for intrinsic lapatinib resistance. We next compared the effects of knocking down TA or TK alone or in combination with HER2 inhibition. While we observed only a minor growth defect upon TA knockdown, breast cancer cells lacking TK cannot proliferate even with intact HER2 signaling, indicating that TK activity is essential for breast cancer cell viability (Supplementary Fig. 4c). This finding is consistent with our initial screening results showing that sgRNA constructs targeting TK are depleted under both vehicle and lapatinib treatment conditions (Supplementary Fig. 4d). TK has also been reported to be essential for colorectal and liver cancer growth[24,28], likely because the TK-catalyzed reactions are non-redundant in the PPP (see Discussion below). In addition, unlike TA, TK expression status does not correlate specifically with HER2-positive breast cancer patient survival (Supplementary Fig. 4e). In sum, these findings suggest that TK functions as an essential enzyme for cell viability in various cell

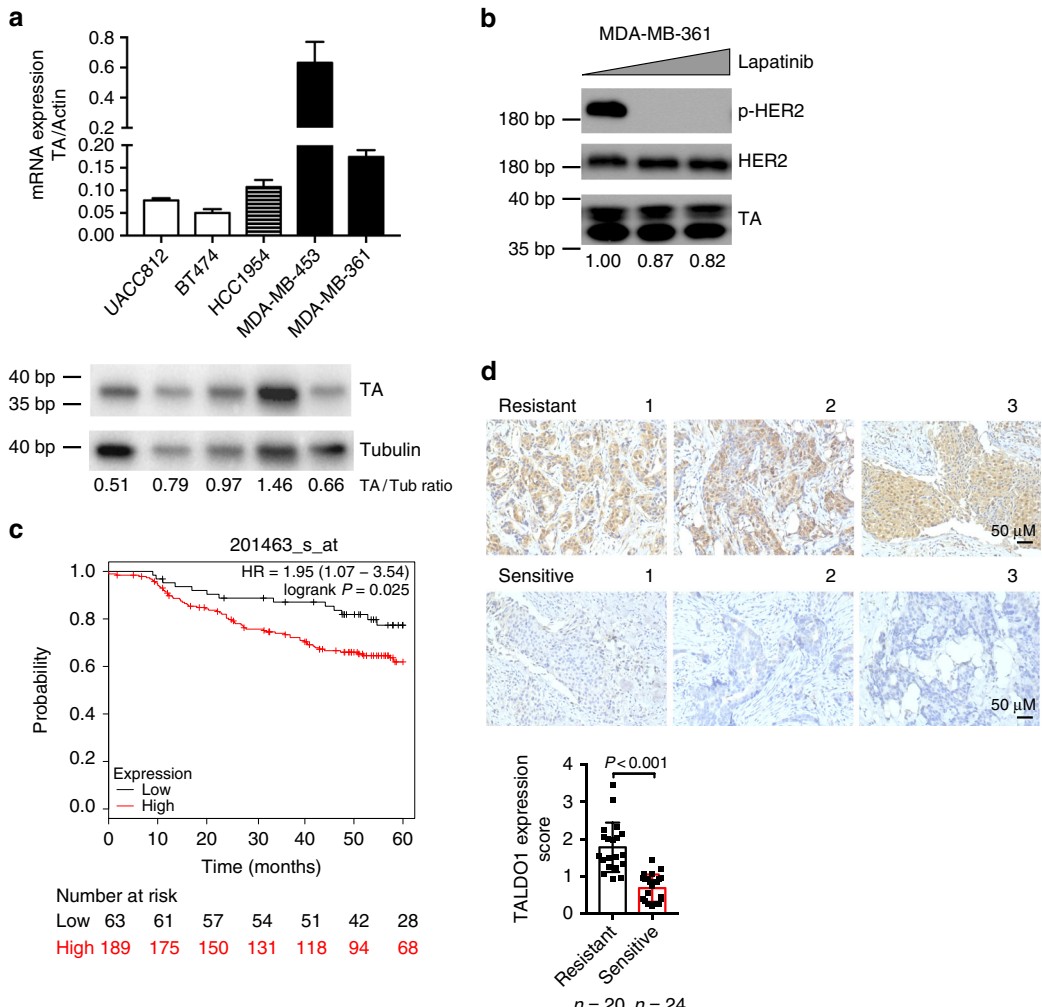

**Fig. 3** TA expression correlates with sensitivity to lapatinib. **a** TA mRNA expression (normalized to actin, top) and protein expression (normalized to tubulin, bottom) of the indicated breast cancer cell lines was evaluated by RT-qPCR and western blotting, respectively. **b** Immunoblotting with indicated antibodies. MDA-MB-361 cells were treated with 0, 1, or 3 µM lapatinib for 72 h and then collected. Band density was quantified using FIJI. **c** Kaplan–Meier relapse-free survival (RFS) analysis of HER2-amplified breast cancer patients stratified by TA expression (TA-low quartile in black, remaining patients in red). **d** Representative TA immunostaining images in resistant (progressive or stable disease) and sensitive (partial remission or complete response) breast cancer tissues. Samples were obtained from HER2-positive breast cancers by core-needle biopsy prior to therapy. $n = 24$ resistant tumors and $n = 20$ sensitive tumors. Quantification of TA immunohistochemistry in resistant and sensitive tumors. Error bars denote the SD, $P$-value calculated with unpaired, two-tail, student $t$-test

types, whereas TA exhibits synthetic lethality only when combined with targeted HER2 inhibition.

**TA is the key enzyme controlling replenishment of PPP metabolic flux.** Several major signaling pathways downstream of HER2 signaling are known to influence cellular metabolism including PI3K/Akt and MAPK[40,41], yet a role for HER2 signaling in regulating the PPP is previously unknown. To investigate this, we conducted metabolic profiling to assess metabolic changes upon HER2 inhibition and identify the mechanism of TA deficiency-induced synthetic lethality. Because the non-oxidative PPP is bidirectional, there are three main directions of non-oxidative PPP metabolic flux: (1) from glycolytic intermediates to generate R5P for nucleotide synthesis; (2) from R5P to generate glycolytic intermediates for energy production; and 3) from R5P to glycolytic intermediates to fuel gluconeogenesis and replenish the oxidative PPP for NADPH generation (arrows in Supplementary Fig. 4a). In order to understand which of these scenarios

predominates in HER2-inhibited breast cancer cells, we performed metabolic labeling using 1,2-$C^{13}$-glucose for 24 h (Fig. 4a).

First, we analyzed the total levels of glycolytic and PPP metabolites to investigate the metabolic alteration induced by combined HER2 inhibition and TA deficiency (Supplementary Fig. 5). Glycolytic flux was significantly inhibited by lapatinib at the step of glucose phosphorylation mediated by hexokinase, while shTA did not affect the first several steps of glycolysis. This is understandable as hexokinase activity is regulated by Akt, a downstream mediator of HER2 signaling[42]. The levels of PPP metabolites were significantly altered by either lapatinib treatment or TA deficiency. Surprisingly, the levels of two TA-specific metabolites, S7P and E4P, increased several-fold under shTA conditions and further increased upon combination treatment with lapatinib, suggesting synergistically altering non-oxidative PPP flux may be necessary for cell death.

1,2-$C^{13}$-labeled glucose has been a useful tool for dissecting PPP metabolic flux from glycolysis[43]. The first $C^{13}$-labeled carbon in glucose is oxidized upon entering the oxidative PPP, resulting

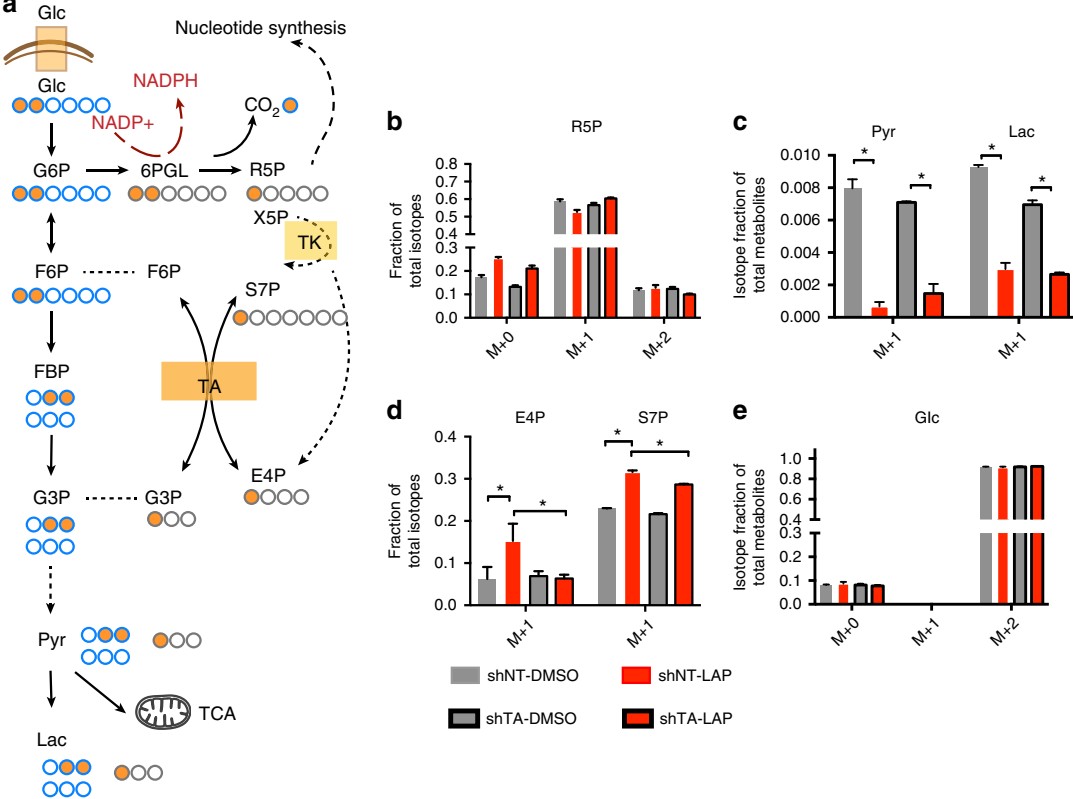

**Fig. 4** The TA-mediated non-oxidative PPP replenishes the oxidative PPP. **a** Schematic depicting the fate of labeled [13]C (orange-filled) from 1,2-[13]C$_2$-labeled glucose in glycolytic [51] and PPP (gray) intermediates. Abbreviations: Glc, glucose; G6P, glucose-6-phosphate; 6PGLU, 6-phosphogluconate; R5P, ribulose-5-phosphate; F6P, fructose-6-phosphate; FBP, fructose-1,2-biphosphate; G3P, glyceraldhyde-3-phosphate; S7P, sedoheptulose-7-phosphate; E4P, erythrose-4-phosphate; Pyr, pyruvate; Lac, lactate. **b-e** Normalized peak areas and distributions of isotopomers of the indicated metabolites. MDA-MB-361 cells were treated with 4 μM lapatinib or DMSO for 24 h, at which point 1,2-[13]C$_2$-glucose supplemented media with DMSO or lapatinib were added and cells were treated for another 24 h before sample collection. *$P < 0.01$ by student $t$-test, error bars denote the SD ($n = 3$).

in M + 1 labeling in R5P and subsequent metabolites such as S7P, E4P and G3P. Alternatively, if R5P is produced via the non-oxidative PPP, it will still carry the 1,2-C[13]-labeling on the M + 2 position. M + 0 R5P comes from the original pool without C[13]-labeling. We first confirmed that, lapatinib treatment significantly inhibits the generation of R5P, as indicated by an increased fraction of M + 0. Moreover, the non-oxidative arm did not support R5P generation, as the M + 1 fraction of R5P did not increase (Fig. 4b). Next, lapatinib treatment inhibited non-oxidative PPP flux toward downstream glycolytic intermediates because M + 1 pyruvate and M + 1 lactate also decreased with treatment (Fig. 4c). Notably, this contrasts with a previous study suggesting that the PPP can be utilized by cancer cells as an alternative pathway to circumvent glycolysis blockade[43]. Instead, our results indicate that lapatinib treatment activates the non-oxidative PPP to fuel gluconeogenesis, thereby replenishing the oxidative PPP, potentially to facilitate NADPH generation (scenario 3 in Supplementary Fig. 4a and Supplementary Fig. 6). Further evidence for this notion is provided by the finding that the M + 1 fraction of the two PPP-specific TA substrates, E4P and S7P, increased with lapatinib treatment and suppression of TA partially blocks this increase (Fig. 4d), suggesting that lapatinib activates this TA-dependent pathway. Finally, HER2 inhibition did not affect glucose uptake in this model as M + 2 fractions did not change among the four groups analyzed (Fig. 4e). Together, these metabolic profiling results indicate that the non-oxidative PPP supports metabolic flux to replenish the oxidative PPP and TA is a central enzyme that controls the efficiency of this pathway.

**TA suppression reduces NADPH and promotes cell death.** Since a major function of the PPP is to generate NADPH for anabolic cellular processes, we directly measured changes in cellular NADP+ and NADPH levels after HER2 inhibition and TA knockdown. Although lapatinib treatment caused a measurable decrease in cellular NADPH, this reduction was strongly exacerbated as a result of TA depletion (Fig. 5a and Supplementary Fig. 7a). Whereas cellular NADP+ was not affected by HER2 inhibition or TA knockdown (Supplementary Fig. 7b–c), the NADP+/NADPH ratio increased significantly in cells with combinatorial TA with HER2 inhibition due to deficient NADPH production (Fig. 5b). Based on these results, we conclude that NADPH levels are largely dependent on TA-mediated PPP activity under HER2 blockade. Known functions of NADPH include reducing oxidative stress and supporting the biosynthesis of various metabolites including fatty acids and nucleotides[23]. Consistent with the previously reported ability of HER2-targeted inhibitors to induce oxidative stress[44,45], we found that levels of the antioxidants NADH and GSH were diminished upon HER2 or TA suppression, with the lowest levels resulting from combinatorial targeting of both pathways (Fig. 5c, d). Because synthesis of palmitate from acetyl-CoA reflects cellular fatty acid synthesis, we used metabolically labeled [13]C- palmitate as a surrogate for lipid synthesis. Like antioxidant levels, labeled palmitate was markedly decreased by either lapatinib treatment or TA knockdown alone and undetectable in the combination group (Fig. 5e). A similar trend was observed for nucleotide synthesis as measured by total levels of IMP, GMP, and AMP (Fig. 5f). Together, these results indicate that, when HER2 signaling is blocked, the

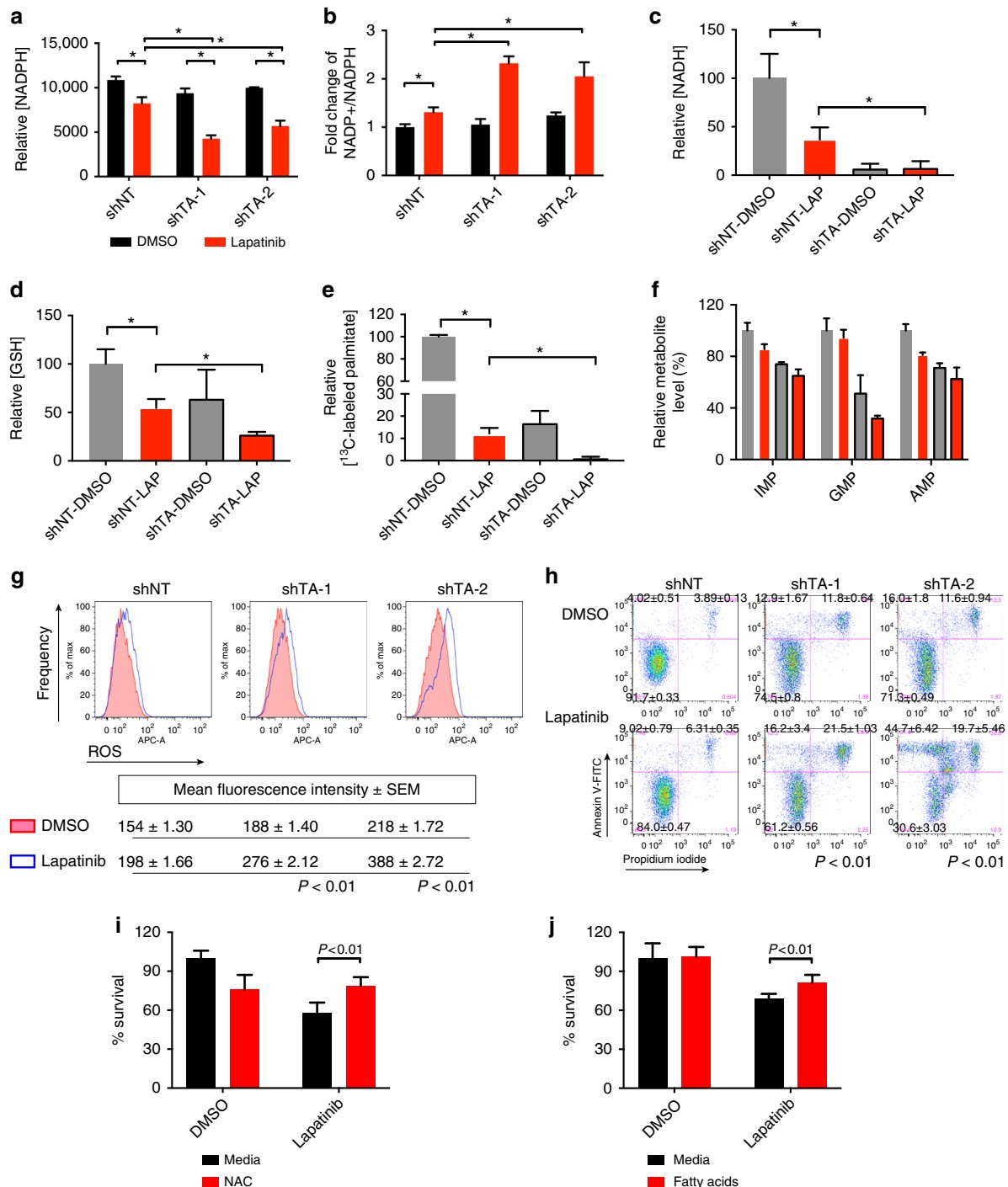

**Fig. 5** TA-dependent NADPH production supports fatty acid synthesis and ROS scavenging. **a**, **b** Relative concentrations of NADPH and fold changes of NADP + /NADPH ratios were detected using a bioluminescent assay ($n = 3$). **c**, **d** Normalized mass spectrometric peak areas of the indicated metabolites. ($n = 3$) **e**, **f** Fraction of M + 2n labeled palmitate, or total IMP, GMP and AMP levels were detected by mass spectrometry-based metabolic profiling ($n = 3$). **g** Cellular ROS levels were measured by Cell Rox deep red flow cytometry staining. Mean fluorescence intensity ± SEM was calculated by FlowJo ($n = 3$). **h** Percentages of apoptotic cells were detected by Annexin V-FITC and PI staining. Numbers indicate the percentages of total evaluated cells. MDA-MB-361 cells were treated with DMSO or 4 μM lapatinib for 48 h in all assays. Numbers showing percentage of each population ± SEM, *P*-value shows Q1 and Q2 percentages with student *t*-test. ($n = 3$). **i**, **j** Relative siTA MDA-MB-361 cell survival with or without 3-day supplementation with NAC (**i**) or fatty acids (**j**) combined with DMSO or lapatinib, as indicated ($n = 5$). *$P < 0.01$, student *t*-test, error bars denote the SD

non-oxidative PPP activity is required for oxidative stress management, fatty acid production, and nucleotide synthesis in order to sustain breast cancer cell growth and viability.

TA-mediated PPP flux is the major cellular source of NADPH and therefore participates in reducing cellular ROS[46]. Therefore,

we directly measured cellular ROS using flow cytometry. As expected based on the reduced levels of NADH and GSH (Fig. 5c, d), we observed an increase in cellular ROS following lapatinib treatment, which was further enhanced upon TA knockdown (Fig. 5g and Supplementary Fig. 7d). Together with the metabolic

profiling evidence described above, we conclude that HER2 inhibition increases cellular ROS by suppressing the oxidative PPP.

Because high levels of ROS are known to induce cell death[47], we next measured apoptosis by co-staining breast cancer cells with propidium iodide and annexin V. Whereas lapatinib treatment and TA knockdown individually slightly increased apoptosis rates in MDA-MB-361 and MDA-MB-453 cells, combination treatment significantly heightened cell death (Fig. 5h and Supplementary Fig. 7e), which likely reflects the elevated levels of cellular ROS (Fig. 5g and Supplementary Fig. 7d). To further dissect which of these metabolites are essential for cell survival, we performed rescue experiments by adding the ROS scavenging agent N-acetylcysteine (NAC), free fatty acids, or nucleosides directly into the culture media of cells treated with combined TA and HER2 inhibition. While supplementation with NAC or fatty acids increased cell survival in the combination group (Fig. 5i–j), nucleosides failed to significantly impact the lapatinib/siTA-induced growth deficit (Supplementary Fig. 7f). Together, these results indicate that inhibiting TA and HER2 activity is synthetically lethal due to increased cellular ROS and decreased fatty acid production which result from reduced levels of cellular NADPH.

## Discussion

In this study, we applied CRISPR/Cas9-based loss-of-function genetic profiling to discover molecules that drive intrinsic resistance to HER2 inhibition in HER2-positive breast cancers. The metabolic enzyme TA, which resides in the non-oxidative PPP, was uncovered from this screen. The essentiality of the PPP for cancer development has been described in recent studies, most of which have concentrated on the role of TK[24,28]. Consistent with those studies, we also observed that suppression of TK reduces cancer cell proliferation, such that TK is an essential metabolic enzyme for HER2-positive breast cancer cells. In contrast to TK, suppression of TA does not affect cancer cell growth but instead synergizes with HER2 inhibition to cause cell death in breast cancer cells resistant to anti-HER2 treatment alone.

These differences in the essentiality of TK and TA might be rationally explained by close inspection of their respective positions within the PPP (Supplementary Fig. 4a). Whereas TK activity is required to generate R5P via the non-oxidative pathway, TA is dispensable in this process as xylulose-5-phophate can also be converted into R5P by an alternative route involving the sequential activity of phosphopentose epimerase and phosphopentose isomerase[23]. As a result, TK activity is indispensable in many types of cancer cells and is likely critical for the growth of non-transformed cells as well, whereas TA activity is only required for proliferation in the presence of cellular stress caused by HER2 inhibition. Based on these considerations, targeting TA might be predicted to uniquely synergize with HER2 inhibition in breast cancer, with reduced toxicity in comparison to blockade of TK, which is independently necessary for mammalian cell growth.

As might be expected based on TA's known function in NADPH production, we also observed increased cellular ROS in cells deficient in TA. HER2 inhibition itself increases cellular ROS by various reported mechanisms which are not yet fully understood. Interestingly, redox-related genes are among the most induced cohorts in an acquired lapatinib resistance model[29], indicating that a cell's capacity to reduce ROS is vital to endure HER2 inhibition. Our study not only confirms the importance of ROS, but elucidates the key function of the PPP in generating NADPH that can be used by cells to reduce oxidative stress or synthesize fatty acids. In lapatinib-resistant breast cancer cells, HER2 inhibition decreases glycolysis

and TCA cycle activity (Supplementary Fig. 5), but decreased flux through these metabolic pathways alone is insufficient to significantly impact cell survival. However, a further deficiency in NADPH production caused by TA depletion induces cell death. In addition to oxidative stress, we also observed a strong impairment in palmitate synthesis upon combinatorial targeting of HER2 and TA. Previous studies have also highlighted the importance of fatty acid synthesis for breast cancer cell viability[48].

Even with its remarkable success, a substantial fraction (20%) of HER2-positive breast cancer patients still fail to respond to the most advanced HER2-targeted therapies[11]. Biomarkers for predicting response to HER2 inhibitors are largely based on previously identified acquired resistance mechanisms, including p95HER2, PTEN, and the estrogen receptor[49]. However, because expression of these proteins typically increases only as a result of selective pressure caused by HER2 inhibitor treatment, these biomarkers are unlikely to accurately identify initial non-responders. Our results indicate that pre-treatment intratumoral TA expression status effectively stratifies patients who will or will not respond to trastuzumab (Fig. 3c, d), rendering it potentially useful as a biomarker, although further analysis with larger patient cohorts is needed. This notwithstanding, our results reveal TA as a metabolic enzyme that, when depleted in combination with HER2 inhibition, exhibits unusual synthetic lethality in breast cancer cells intrinsically resistant to therapeutic approach alone.

## Methods

**Cell lines and reagents**. All breast cancer cell lines used in this study were obtained from the Duke Cell Culture Facility (originally from ATCC). MDA-MD-361, MDA-MD-453, AU565 and BT-474 cells were cultured in RPMI1640 supplemented with 10% FBS, 2 mM glutamine, 10 mM HEPES, 1 mM sodium pyruvate, 2.5 g/L glucose and 1% penicillin-streptomycin (Thermo Fisher Scientific). UACC812 cells were cultured in DMEM supplemented with 10% FBS and 1% penicillin-streptomycin. rBT474 cells were established as previously reported[38], whereas rAU565 cells were developed by 2-month selection with 1 μM lapatinib. These acquired resistant lines were maintained in 1 μM lapatinib added to regular media with supplements. All cell lines were grown at 37 °C with 5% CO$_2$. Mycoplasma contamination was examined using Lonza MycoAlert kit (LT07). Lapatinib (L-4899) was purchased from LC Laboratories. Oxythiamine (O4000) was purchased from Sigma-Aldrich. NAC (Sigma A7250, 500 mM), fatty acids (Sigma F7050, 1:1000), or nucleosides (Millipore ES-008-D, 1:100) were incubated with the indicated cell lines for 3 days.

**CRISPR/Cas9 library amplification**. The CRISPR/Cas9 library was previously established in the Wood Lab[33]. Library plasmid amplification was modified from a previous report[30]. Briefly, 200 ng of library plasmid was added into 50 μL competent cells (Stratagene 200314), following standard protocols, and then seeded onto ampicillin (100 μg/mL) LB agar plates for 16-h growth. Twenty replicates were performed to yield a coverage of 2000x for each sgRNA. Colonies were scraped off plates and combined before plasmid DNA was extracted using Endotoxin-Free Midi Prep (Qiagen 12943).

**Lentivirus production and titering**. HEK293T cells were seeded on 10 cm plates and grown overnight to approximately 70% confluence at the time of transfection. A total of 20 μg of plasmids were transfected using 60 μL Lipofectamine 2000 (Thermo Fisher Scientific 11668019) following standard procedures. A ratio of 5:4:1 (lentiCRISPR library plasmid: psPAX2: pVSVg) was used. After 6 h, the media was changed to 12 ml RPMI1640. After 48 h, viral particles were collected, filtered (0.45 μM), aliquoted, and frozen at −80 °C for short-term storage.

To test virus titers, MDA-MB-361 cells were seeded onto 96-well plates at 2000 cells/well overnight. Ten ratios of virus media/fresh media from 1:40-1:1 were added into wells (four replicates each) for 24 h, followed by 48 h puromycin selection (2 μg/mL). Cell survival was determined by Cell-Titer-Glo (Promega G7570). The dose of 20% survival, which reflects a MOI of 0.2 was used for screening.

**Pooled screening using the CRISPR/Cas9 library and analysis**. A total of 20 million MDA-MB-361 cells were seeded into two 15 cm plates, infected with virus at MOI 0.2 for 24 h, and then selected with puromycin for 2 days, at which point samples were collected to assess library representation. Cells were maintained at 1,000x library coverage in puro for 7 days to allow for the generation of knockout cells, then split into four biological replicates (two of each treated with DMSO or

lapatinib) for 2 weeks. After genomic DNA was extracted using a DNA extraction kit (Qiagen 69504), two rounds of PCR were performed to amplify the sgRNA insertion cassette[30].

Purified PCR products were sequenced with a read length of paired-end 101 bp on Illumina HiSeq3000sequencer. There are more than 3 million reads for each sample. The sequenced reads were aligned to the sequence library of single-guide RNAs to count each target in different samples and perform statistical test among conditions using the MAGeCK(v0.5.3) algorithm with default settings[50]. For each sgRNA construct, the frequency (FR) was calculated as sgRNA reads/total reads of the sample. The relative depletion of each sample was calculated as the (FR of sample) / (FR at $T = 0$). The sensitivity relative depletion percentage was calculated as the (FR of lapatinib)/(FR of DMSO).

**Crystal violet staining**. Crystal violet staining solution consisted of 10% methanol with 0.1% crystal violet powder (Sigma-Aldrich C3386). Breast cancer cells were seeded in 2 cm plates overnight, then treated with DMSO or lapatinib for 72 h. Cells were then washed 2 times with PBS, stained for 30 min at room temperature, washed 3 additional times with PBS, and finally air-dried for 1 h before photographic analysis.

**shRNA and sgRNA constructs**. sgRNA constructs were cloned using an established lentiCRISPRV2 GeCKO protocol (http://genome-engineering.org/gecko/). Forward string sequences were as follows:

sg*TALDO1*-1: CAAGCAGTTCACCACCGTGG;
sg*TALDO1*-2: GGAAAGACTTCTCATCCAGG;
sg*IGF1R*-1: GGGACCAGTCCACAGTGGAG;
sg*IGF1R*-2: GAGGGGTTTGTGATCCACGA;
sg*GATA3*-1: GGGGTGGTGGGTCGACGAGG;
sg*GATA3*-2: GCAGTACCCGCTGCCGGAGG;
sg*TBX3*-1: GGAGCCCGAAGAAGAGGTGG;
sg*TBX3*-2: CGAGGGTGAGAGCGACGCCG;
sg*TKT*-1: CATCCAGGCCACCACTGCGG;
sg*TKT*-2: CAAGGGCAGGATCCTCACCG.

shTA constructs were purchased from Sigma-Aldrich: shTA1 (TRCN0000052520) CCGGCTGCAACATGACGTTACTCTTCTCGAGAAGAGTAACGTCATGTTG-CAGTTTTTG; shTA2 (TRCN0000052521) CCGGGCGGATGCTGACAGAACGAATCTCGAGATTCGTTCTGTCAG-CATCCGCTTTTTG.

**siRNA sequences and transfection**. siNT and siTA pooled constructs were dissolved in nuclease free water to 10 μM, then aliquoted and stored in −20 °C. For transfection, 2.5 pM siRNA and 0.25 μL DharmaFECT were added to each well in a 96-well format. After 24 h, fresh media replaced the transfection mix and subsequent cell treatment were performed after 48 h as indicated. siRNA constructs were purchased from Dharmacon. Target sequences are as follows. siNT: UGGUUUACAUGUCGACUAA; siTA (SMART pool): (1) GCAAACACCGA-CAAGAAAU, (2) UCACAAGAGGACCAGAUUA, (3) CCGAGUAUCCACA-GAAGUA, (4) ACAAGAAGUUUAGCUACAA.

**Cell growth inhibition assays**. Cells were seeded in 96-well plates at 4000 cells/well overnight, then treated with different doses of lapatinib as indicated and/or other indicated compounds. Cell-Titer-Glo reagent (Promega G7570) was added into each well according to the manufacturer's protocol to measure cell viability.

**Western blotting and antibodies**. Total protein was extracted using lysis buffer containing 0.9% NP-40, 1 mM EDTA, 50 mM Tris-HCl (pH 8.0), 50 mM NaCl (NETN) supplemented with 1% protease inhibitor cocktail (Thermo Fisher Scientific 78430). Lysates were first sonicated and centrifuged at 12,000 r.p.m. for 10 min. Protein concentrations were determined by BCA assay (Thermo Fisher Scientific 23227). Supernatants were then collected and boiled in SDS loading buffer for 10 min. A total of 15 μg of protein samples were loaded on SDS/PAGE gels and transferred to nitrocellulose membranes (Millipore-Billerica). Membranes were blocked in 5% bovine serum album in Tris-buffered saline with 0.1% Tween-20 (TBST) and probed with the following antibodies: TA (Proteintech Group 12376-1-AP, 1: 2000), tubulin, phospho-HER2, and HER2 (Cell Signaling 3873 1:5000, 2243 1:1000, and 4290 1:1000). Secondary antibodies (Life Technologies G21040 1:10,000and G21234 1:10,000) were visualized using a chemiluminescent reagent Pico kit (Thermo Fisher Scientific 35350). Protein ladder (Thermo Fisher 26616) was used to identify molecular weight. Uncropped blots are shown in Supplementary Fig. 8.

**Quantitative RT-PCR analysis**. Total RNA was extracted using RNease mini kit (Qiagen 74104) and converted to cDNA with iScript cDNA synthesis kit (Bio-Rad). PCR was performed on a MasterCycler RealPlex4 real-time PCR system (Eppendorf) using specific primer pairs for the indicated genes. Primer sequences are shown below.

TA fwd: ATCCTGGGGCTTGTACTCGT; rev: GAAGCGTCAGAGGATGGA GT.

TK fwd: GCATGGTGTGGAAAAAGAGG; rev: CGCCTACGTATCAGCTC CA.

G6PD fwd: CACCAGATGGTGGGGTAGAT; rev: AGAGCTTTTCCAGGGCG AT.

PGD fwd: GCCTTGGAAGATGGTCTTGA; rev: GTCAGTGGTGGAGAGGAAGG.

**Metabolite extraction**. Metabolites were extracted directly from cultured cells after treatment. After snap-freeze in liquid nitrogen, 500 μL ice-cold 80% methanol/water were added into each well in a 6-well plate and spun down at 20,000 r.c.f. for 10 min, also described in a previous paper[51]. Supernatants were split and transferred to two new Eppendorf tubes (one for back up) and dried using a vacuum concentrator at room temperature. Dry pellets were reconstituted in 30 μL solvent (water:methanol:acetonitrile, 2:1:1, v/v/v) and 3 μL was analyzed by liquid chromatography-mass spectrometry (LC-MS).

**LC-MS**. Ultimate 3000 UHPLC (Dionex) coupled to a Q Exactive Plus-Mass spectrometer (QE-MS, Thermo Scientific) was used for metabolite profiling. A hydrophilic interaction chromatography method (HILIC) employing an Xbridge amide column (100 × 2.1 mm i.d., 3.5 μm; Waters) was used for polar metabolite separation. LC was performed as previously described[52], except that mobile phase A was replaced with water containing 5 mM ammonium acetate (pH 6.8). The QE-MS was equipped with a HESI probe with related parameters set as below: heater temperature, 120 °C; sheath gas, 30; auxiliary gas, 10; sweep gas, 3; spray voltage, 3.0 kV for the positive mode and 2.5 kV for the negative mode; capillary temperature, 320 °C; S-lens, 55; scan range ($m/z$): 70 to 900 for pos. mode (1.31 to 12.5 min) and neg mode (1.31 to 6.6 min) and 100 to 1000 for neg. mode (6.61 to 12.5 min); resolution: 70,000; automated gain control (AGC), $3 \times 10^6$ ions. Customized mass calibration was performed prior to data acquisition.

**Metabolomic data analysis**. LC-MS peak extraction and integration were performed using commercially available software (Sieve 2.2, Thermo Scientific). Peak area was used to represent the relative abundance of each metabolite in each sample. Missing values were handled as previously described[52]. Fractions of each isotopes were corrected of nature abundance using the IsoCor software[53].

**Quantification of cellular NADP+ and NADPH levels**. Cellular NADP + and NADPH levels were measured using NADP/NADPH Glo Assay (Promega G9081). For each condition, four thousand cells with three biological repeats were seeded, treated with DMSO or lapatinib for 72 h, and then collected for the assay following the manufacturer's protocol.

**Quantification of ROS and apoptosis**. Cells were seeded overnight in 12-well plates at 20,000 cells per well, and then treated with DMSO or lapatinib for 48 h. CellRox Deep Red (Thermo Fisher Scientific C10422) was added at 1:1000 into media for 30 min, after which cells were collected and washed 2 times with FACS buffer (1% FBS in PBS). For apoptosis assays, cells were harvested and immediately stained with Alexa Fluor 488 Annexin V/Dead Cell Apoptosis Kit (Thermo Fisher Scientific V13241). Flow cytometry was performed in the Duke Cancer Center shared flow cytometry facility.

**Patient sample collection and analysis**. Primary invasive ductal carcinomas of the breast with HER2 positivity were obtained from 44 female breast cancer patients before and after pre-operative neoadjuvant therapy in the Breast Tumor Center, Sun-Yat-Sen Memorial Hospital (SYSMH), Sun-Yat-Sen University between January 2015 and December 2016. All patients underwent 4–8 cycles of neoadjuvant chemotherapy with taxanes and trastuzumab-based regiment with or without anthracycline (TCbH or EC followed by TH) according to NCCN guideline. Primary diseases were measured or evaluated using clinical and radiological methods in accordance with Response Evaluation Criteria in Solid Tumor (RECIST). Breast tumor samples were obtained via core-needle biopsy prior to treatment. The collected tumor tissues were embedded for studies of histology, HE staining and immunohistochemistry. All samples were collected with informed consent according to the internal review and ethics boards of the hospital.

HER2 status was determined by IHC or Fish. TA expression was examined by immunohistochemistry on paraffin-embedded tissue sections. Briefly, anti-TA (GeneTex, GTX102076, 1:100) was used as the primary antibody for overnight incubation at 4 °C. Sections were subsequently treated with goat anti-rabbit secondary antibody (1:100, CST7074, Cell Signaling Technology), followed by further incubation with streptavidin- horseradish peroxidase complex. Diaminobenzidine (ZSGB-BIO, ZLI-9017,) was used as a chromogen and sections were lightly counterstained with hematoxylin. Cytoplasmic and nuclear TA staining were scored using a modified H-Score method. Briefly, H-Score is obtained by the formula 4 × % of strongly staining cells + 3 × % of moderately to strongly staining cells + 2 × % of moderately staining cells + 1 × % of weakly staining cells,

giving a range of 0–4. The percentage of positively stained tumor cells was calculated at all fields per section as evaluated at ×400 magnification.

**Statistical analysis**. Unless otherwise specified, unpaired, two-tailed student's *t*-tests using PRISM were performed and *p* values are noted in each figure legend. Results are presented as means ± SEM or SD in legends.

## Data availability

All relevant data are available within the manuscript and its supplementary information or from the authors upon reasonable request. Please contact Xiao-Fan Wang (xiao.fan.wang@duke.edu) or Yi Ding (ding.yi@duke.edu) for any requests. The KM-plot data is from website: http://kmplot.com/analysis/.

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

## Acknowledgements

The authors thank Grace Anderson and Peter Winter for development of the CRISPR/Cas9 screening library and for advice on experimental procedures; Chen Jin for help with sequencing and read mapping; Xiao-Jing Liu, Juan Liu, and Jason Locasale for metabolic profiling, data analysis and discussion; Sarah Sammons for discussion on HER2-targeted therapies in clinical use; and James Alvarez for helpful discussion. This work was supported by a National Key R&D Program of China 2017YFC1309103 to C.G.; a DOD grant W81XWH-16-1-0618 to X.-F.W.; a DOD grant W81XWH-16-1-0703 to K.C.W.; CA190991 from the NIH to Q.-J.L., 5F30CA206348 from the NIH to K.H.L.

## Author contributions

Conceptualization and Methodology, Y.D., R.C., D.H., Q.-J.L., X.-F.W. and K.C.W.; Investigation—cell culture, screening and hits validation, Y.D., L.Y, H.X., T.Y. and J.C.; Investigation—sequencing, read mapping and analysis, Y.D. P.S., K.H.L. and Q.-F.W; Investigation—Human data acquisition and analysis, C.G., G.L., E.-W.S.; Writing— Original Draft, Y.D.; Writing—Reviewing & Editing, P.B.A., K.C.W. and X.-F.W.; Supervision and Funding Acquisition, C.G. K.C.W. and X.-F.W.

## Additional information

**Competing interests:** The authors declare no competing interests.

