## [Peer Review File · Nature Communications]

Reviewers' comments:

Reviewer #1 (Remarks to the Author):

In their manuscript, Ding et al. describe that HER2 inhibition resistant breast cancer cell lines are sensitive to genetic targeting of transaldolase (TA), an enzyme in the non-oxidative pentose phosphate pathway. The authors identify TA, but not TK, as synthetic lethal with lapatinib treatment using a CRISPR-Cas9 library that targets a focused set of signaling and metabolic genes. Their experiments suggest that TA depletion reduces NADPH levels and lipid/nucleotide synthesis. Finally they showed a correlation between high TA expression and poor response to HER2 inhibitors. There are some issues that needs to be addressed, in particular a more detailed mechanistic explanation for why TA is required under lapatinib treatment.

Major points:

- The authors claim in the manuscript that intrinsic and acquired resistance after long-term treatment with the drug depends on completely different factors. While this may be true, the authors should provide further experiments to support this statement. (i.e. using at least one acquired-resistance model to lapatinib and test if TA gene disruption would also sensitize the cells to the drug.)
- Does TA knockdown in responsive cell lines sensitize them to lapatinib treatment at much lower concentrations of the drug?
- Resistant cells depend on high NADPH levels to support growth under lapatinib treatment, likely for ROS, nucleotide synthesis or just de novo lipogenesis. The authors should precisely determine the reason for this NADPH change. Can they rescue cell death using antioxidants (i.e NAC or trolox?), nucleosides in the presence of lapatinib, in order to dissect the limiting metabolic factor.
- Metabolite profiling experiments seem to partially support that TA helps to redirect the carbons from non-oxidative PPP to ox-PPP in order to produce NADPH. However, some extra interpretation of the data would be helpful. Do the authors have a hypothesis of why lapatinib treatment significantly decrease levels of all the ox-PPP intermediates even in shNT cells? (Fig. S4A).

Minor:

- In Figure 1E, the authors should use the same scale for the 2 graphs to actually see the real difference in cell growth and how significant it is.
- TA mRNA levels increase in intrinsically resistant cell lines compared to sensitive ones (Fig. 2A). However, one should really assess whether TA is increased at the protein level (i.e. a TA western blot of resistant (as many as possible) vs sensitive (as many as possible) cell lines).

Reviewer #2 (Remarks to the Author):

The study by Ding and colleagues describes a novel synthetic lethal interaction between HER2 inhibition and the metabolic enzyme transaldolase. The authors identified transaldolase in a focused CRISPR KO screen of 378 signaling and metabolic genes as a synthetic lethal partner of the HER2 inhibitor lapatinib in MDA-MB-361 cells. The author showed that transaldolase expression is associated with poor prognosis in HER2-amplified breast cancer patients, and transaldolase expression is elevated in tumors that are resistant to treatment. The authors carried out extensive metabolic labeling experiments to identify changes in the pentose phosphate pathway that are associated with HER2 inhibition and transaldolase inhibition that might explain the observed synthetic lethality. Overall the experiments are well designed and the data is of high quality, and the manuscript is clearly written. The connection between transaldolase and HER2 inhibitor resistance is novel and could be of important prognostic value. The study suggests that transaldolase to be a potential target for synergizing with HER2 inhibitors in patients that show intrinsic resistance to single agent lapatinib.

Major points

1. A major concern with this study is that the authors primarily used a single breast cancer cell line, MDA-MB-361, for mechanism study. This cell line harbors HER2 amplification but is intrinsically resistant to lapatinib. CCLE data indicates that this cell line harbors a large number of mutations, including BRAF(V600E) and PIK3CA(E545K). Both these mutations could explain the cell line's resistance to lapatinib. Thus, to demonstrate transaldolase to be an independent factor in lapatinib resistance and a general synthetic lethal partner of HER2 inhibitors, it would be important for the authors to demonstrate the same mechanism in additional HER2-amplified cell lines that exhibit intrinsic lapatinib resistance, preferably without known resistance mechanisms such as those mentioned above. Alternatively, would transaldolase expression in an lapatinib sensitive cell line such as BT474 confer resistance to the drug?
2. Mechanistically, the authors postulated that transaldolase knockdown could perturb three pathways that synergize with lapatinib: ROS elevation, deficient lipid synthesis, and reduced nucleotide synthesis. Some of these hypotheses could be tested by chemical rescue experiments in vitro. Can the synthetic lethal mechanism between TA and lapatinib be rescued by treating cells with media supplemented with N-acetyl cysteine, free fatty acids such as palmitate and/or cell permeable nucleosides?

Minor points

1. For the sgRNA screen, both DMSO and lapatinib treated cells were cultured for 2 weeks. How many population doublings were there for each arm?
2. For Figure 1C, it would be more informative to plot the average of the two replicas for the DMSO treated samples vs. lapatinib treated samples. This will give a better illustration for which genes are straight lethal and which ones only dropout from lapatinib treated samples. Similarly, for Figure 1D, Figure S1D-G, they should be re-plotted in the format as Figure S3E.
3. Figure S2, it would be useful to indicate the IC50 values for each condition.
4. Figure 2B showed that HER2 phosphorylation was completely inhibited by lapatinib at 1uM. Some experiments in the paper were done at either 4uM or 10uM of lapatinib. This raises the concern that whether the higher concentrations of lapatinib used could introduce off-target effect on other tyrosine kinases such as IGF1R. The author should be able to observe the same synthetic lethal effect at 1uM of lapatinib, or alternatively, with trastuzumab as an independent means to inhibit HER2. As a third alternative, HER2 can be inhibited by shRNA knockdown (CRISPR probably won't work since HER2 is amplified and cells might just get killed by Cas9-induced DNA damage).
5. For the K-M curves in Figure 2C and Figure S3F, were these univariate or multivariate analysis? Were the TA low and high patients stage-matched (i.e. no stage skewing in TA high patients)?
6. Figure 3B and Figure S4A, in Figure 3B, M+0 labelled R5P is elevated by lapatinib; but in Figure S4A total R5P level is reduced by lapatinib. Along this same line, in Figure 4F, IMP, GMP and AMP levels are down with lapatinib or with TA knockdown. It is not clear why such differences were seen and whether these experiments measured different flux of ribose and nucleotide synthesis.
7. Figure S4A, it is not clear what 6PGLU is and why it's level is strongly inhibited by either lapatinib or TA knockdown strongly depleted 6PGLU levels. Is this important?
8. Figure 3D, it is not clear whether the induction of E4P and S7P by lapatinib is a known metabolic response that has been previously published? If so what is the mechanism? In addition, it is also unclear how the transaldolase-dependent induction of E4P and S7P upon lapatinib treatment is related to change in NADPH levels in Figure 4.
9. Figure S3A and S3B – the cell lines in the figure legend are reversed? Should the left be MB-361 and the right be BT474? It would be useful to just put cell line names directly above the bar charts for Figure S3A, S3C and S3D.
10. Figure 4G and 4H, it would be useful to provide Mean +/- SEM and p-values for these experiments.
11. Since PPP enzymes are induced in BT474 cells upon lapatinib treatment, does TA knockdown further sensitize this cell line towards low dose lapatinib?

Reviewer #3 (Remarks to the Author):

The authors perform a CRISPR selective vulnerability screen to identify targets that will be synthetic lethal with HER2 inhibition in breast cancer.

The biological question is important, The emergence of CRISPR screening has proven to be superior over previous RNAi based method, thus the paper is timely and of interest to a wide community.

specific comments:

Figure 1C - The description of this figure in the text, x and y labels and the figure labels does not match.

The text refers to genes which the x and y labels refer to sgRNAs, also the legends refer to the ratio of late to early time point, in this case essential genes should have been visible on this plot. I assume that this plot represents the ratio between drug to DMSO.

Not clear why the authors focused only on the five top hits, where only those significant?

Based on figure 1E it appears that TA KD affects also cells without lapatinib, would this affect its utility as a drug target? On the same line, while figure 1C nicely emphasizes the increased effect of specific genes with lapatinib compared to control, it does not provide any information on whether those genes are also essential for growth in general. A plot of the ratio between late to early time point comparing DMSO to lapatinib would help to clarify this.

Figure 1F - does not match the description in the text, the actual figure is not referenced in the text

The authors should show that the synthetic lethality between HER2 inhibition and TA KO/KD does not occur (or to a much reduced extent) in cells that are not resistant to HER2 inhibition

It is not clear why the authors analyzed TA expression between resistance and non-resistance cell lines at the mRNA level and then analyzed if lapatinib affects TA levels at the mRNA levels. It would be useful to show the former using protein levels as well.

Point-by-point response letter

Reviewer #1 (Remarks to the Author):

In their manuscript, Ding et al. describe that HER2 inhibition resistant breast cancer cell lines are sensitive to genetic targeting of transaldolase (TA), an enzyme in the non-oxidative pentose phosphate pathway. The authors identify TA, but not TK, as synthetic lethal with lapatinib treatment using a CRISPR-Cas9 library that targets a focused set of signaling and metabolic genes. Their experiments suggest that TA depletion reduces NADPH levels and lipid/nucleotide synthesis. Finally, they showed a correlation between high TA expression and poor response to HER2 inhibitors. There are some issues that need to be addressed, in particular a more detailed mechanistic explanation for why TA is required under lapatinib treatment.

We appreciate Reviewer #1's precise summary of our study. We have attempted to address all concerns raised by Reviewer #1 in the comments below.

Major points:

1. The authors claim in the manuscript that intrinsic and acquired resistance after long-term treatment with the drug depends on completely different factors. While this may be true, the authors should provide further experiments to support this statement. (i.e. using at least one acquired-resistance model to lapatinib and test if TA gene disruption would also sensitize the cells to the drug.)

We thank Reviewer #1 for this constructive comment. While previous studies using acquired anticancer drug resistance models unveiled several important mechanisms, our genetic screening approach was designed to uncover new synthetically lethal factors for intrinsically resistant HER2-positive breast cancer cells. In order to investigate whether TA is also involved in mediating acquired resistance, we generated TA-deficient cell lines on the background of two independent acquired resistance models: 1) rBT474 cells previously established by Dr. Neil Spector's lab [1]; and 2) rAU565 cells generated by us via two-month lapatinib treatment in culture. Dose-

response curves indicate that TA knockdown does not sensitize either of these cell lines to lapatinib (Fig. R1a-b, also in revised manuscript Fig. 2d and supplementary Fig. 2g). To investigate this further we also compared TA expression levels in the acquired resistant cells to their parental (sensitive) lines. For both models, TA protein abundance was comparable between the parental and resistant cell lines (Fig. R1c). Based on these results, we believe that TA's ability to mediate breast cancer cell resistance to HER2 inhibition is more specific to the intrinsic resistance condition.

Fig. R1 TA deficiency does not sensitize acquired resistant cell lines.

2. Does TA knockdown in responsive cell lines sensitize them to lapatinib treatment at much lower concentrations of the drug?

We used the same method to generate three TA-deficient cell lines reported to be responsive to lapatinib (BT474, AU565 and UACC812)[2] and performed lapatinib dose-response studies at sub-micromolar concentrations. TA knockdown in all three cell lines

did not further sensitize them to lapatinib (Fig. R2, also in revised manuscript Fig. 2d and supplementary Fig. 2f, h). We interpret this to mean that the responsive cell lines are predominantly dependent on HER2 signaling such that inhibiting HER2 results in dysregulated intracellular signaling, metabolism and other processes that are vital for cell survival and proliferation. In this way, the PPP and other major metabolic pathway may have already been significantly influenced by downstream effector molecules of HER2 such that TA deficiency cannot further sensitize these cells to lapatinib.

Fig. R2

Figure R2. TA deficiency does not further sensitize breast cancer cell lines that are sensitive to lapatinib.

3. Resistant cells depend on high NADPH levels to support growth under lapatinib treatment, likely for ROS, nucleotide synthesis or just de novo lipogenesis. The authors should precisely determine the reason for this NADPH change. Can they rescue cell death using antioxidants (i.e NAC or trolox?), nucleosides in the presence of lapatinib, in order to dissect the limiting metabolic factor.

NADPH is a major output of the PPP and its depletion due to TA knockdown can be interpreted based on the metabolic profiling results. Briefly, lapatinib inhibits influx into the ox-PPP, the major source of NADPH production, while TA knockdown suppresses the non-ox-PPP needed to replenish the PPP, resulting in the observed NADPH depletion. (Further interpretation of this metabolic profiling data is provided below in point 4). We agree with Reviewer #1 that metabolic rescue experiments would provide stronger evidence to further validate our observations. To test which NADPH downstream metabolite(s) are essential for cell survival under lapatinib treatment, we used N-acetylcysteine (NAC) to reduce cellular ROS, or provided lipid (Sigma F7050) or nucleoside (Millipore ES-008-D) supplementation to supply cells with free fatty acids or nucleosides, respectively (Fig. R3, also shown in revised manuscript Fig. 5i-h and supplementary Fig. 7f).

The addition of NAC led to the improved survival of TA depleted cells treated with lapatinib. Therefore, we believe that excessive ROS is one of the key factors inhibiting cell growth in lapatinib-treated cells. We also observed a mild rescue effect upon providing free fatty acids to these cultures. This finding is consistent with previous studies indicating that fatty acid synthesis is essential for HER2-positive breast cancer cells [3, 4]. While nucleoside addition did not provide a significant rescue effect under these conditions, it is possible that nucleosides may further stimulate cell growth when combined with fatty acids and/or ROS scavengers. From these results, we conclude that excessive cellular oxidative stress and reduced fatty acid synthesis are consequences of NADPH shortage that underlie breast cancer cell sensitivity to HER2 inhibition.

Fig. R3

Fig. R3 Supplementation of the ROS scavenger NAC or free fatty acids to TA knockdown cells partially rescues breast cancer cell survival.

4. Metabolite profiling experiments seem to partially support that TA helps to redirect the carbons from non-oxidative PPP to ox-PPP in order to produce NADPH. However, some extra interpretation of the data would be helpful. Do the authors have a hypothesis of why lapatinib treatment significantly decrease levels of all the ox-PPP intermediates even in shNT cells? (Fig. S4A).

In order to better visualize the metabolic flow and clarify our interpretation of this data, we added another two supplementary figures to show the landscape of changes in total metabolite levels and in the fraction of metabolite isotopes (Fig. R4 and R5, also shown in supplementary Fig. 5-6).

We hypothesize that lapatinib decreased all ox-PPP intermediates through the inhibition of hexokinase (HK), which is known to be downstream of Akt [5]. This hypothesis is supported by our measurements of the total levels of glucose and G6P. While cellular glucose levels were not changed among all conditions, total levels of G6P decreased significantly with treatment. This result is consistent with inhibition of HK-mediated metabolism of glucose to G6P. Moreover, this reaction was not affected by TA deficiency.

We also noticed that the G6P downstream metabolites in the ox-PPP, 6PGLU and R5P, decreased similarly upon lapatinib treatment, but varied in the combinational treatment group (Fig. R4), confirming that the effect of lapatinib on the ox-PPP is TA-dependent. However, the regulatory mechanisms of enzymes mediating these steps is not well understood. Although the activity of enzymes can be regulated by the concentration of substrate, product or other factors, we speculate that the accumulation of R5P under combined HER2 and TA inhibition might result from accumulation of the metabolic product S7P (Fig. R4).

Fig. R4

Relative Levels of Metabolites

Fig. R4 Total levels of glycolytic and PPP metabolites under the indicated conditions.

The isotopic labeling data is complicated to dissect due to the reversible nature of the non-ox-PPP. In the manuscript, our strategy was to use definitive data to exclude two out of three possible non-ox-PPP metabolic outcomes: 1) the non-ox-PPP fuels the generation of R5P; 2) the non-ox-PPP fuels downstream glycolysis to produce pyruvate for energy production; and 3) the non-ox-PPP flows into gluconeogenesis. R5P M+1 and M+2 represents the contributions from ox-PPP and glycolysis, respectively. The first outcome is not likely because the M+2 fraction of R5P did not increase: if R5P is generated from the ox-PPP, it will be M+1; if generated from the non-ox-PPP (from glycolysis), it will be M+2. The fact that M+1 in pyruvate and lactate did not increase doesn't support the second outcome: if the non-ox-PPP flows towards downstream glycolysis, there should be more M+1 [6]. Therefore, our data is more consistent with the third outcome. We were able to measure NADPH levels in other experiments to confirm this as well.

Fig. R5

The ¹³C-labeling pattern of metabolites in glycolysis and PPP

Fig. R5 Isotopes of glycolytic and PPP metabolites.

Minor:

1. In Figure 1E, the authors should use the same scale for the 2 graphs to actually see the real difference in cell growth and how significant it is.

We thank Reviewer #1 for pointing this out. The scales for these two graphs have been changed as suggested and the new figures are arranged in Fig. 2b in the revised manuscript.

2. TA mRNA levels increase in intrinsically resistant cell lines compared to sensitive ones (Fig. 2A). However, one should really assess whether TA is increased at the protein level (i.e. a TA western blot of resistant (as many as possible) vs sensitive (as many as possible) cell lines).

To assess TA protein expression as it relates to lapatinib responsiveness more thoroughly, we measured protein levels in five independent HER2-positive breast cancer cell lines (Fig. R6a, also in revised manuscript Fig. 3a). While resistant MDA-MB-453 cells express the highest basal level of TA, we could not detect any obvious correlation between basal TA expression and sensitivity. Given the limited number of cell lines available, to better address this concern we collected an additional 24 clinical patient samples (in addition to the previously collected 20 samples) and analyzed TA expression in HER2-positive tumors from patients receiving neoadjuvant trastuzumab plus chemotherapy, in accordance with NCCN guidelines. For these 44 total HER2-positive biopsies analyzed, TA expression in unresponsive (resistant) patient tumors was significantly higher than in responsive (sensitive) ones (Fig. R6b, also in revised manuscript Fig. 3d). This result demonstrates that TA expression correlates with breast cancer patient response to HER2 inhibition.

Fig. R6

Fig. R6 TA expression in HER2-positive cell lines and breast cancer patient biopsies.

Reviewer #2 (Remarks to the Author):

The study by Ding and colleagues describes a novel synthetic lethal interaction between HER2 inhibition and the metabolic enzyme transaldolase. The authors identified transaldolase in a focused CRISPR KO screen of 378 signaling and metabolic genes as a synthetic lethal partner of the HER2 inhibitor lapatinib in MDA-MB-361 cells. The author showed that transaldolase expression is associated with poor prognosis in HER2-amplified breast cancer patients, and transaldolase expression is elevated in tumors that are resistant to treatment. The authors carried out extensive metabolic labeling experiments to identify changes in the pentose phosphate pathway that are associated with HER2 inhibition and transaldolase inhibition that might explain the observed synthetic lethality. Overall the experiments are well designed and the data is of high quality, and the manuscript is clearly written. The connection between

transaldolase and HER2 inhibitor resistance is novel and could be of important prognostic value. The study suggests that transaldolase to be a potential target for synergizing with HER2 inhibitors in patients that show intrinsic resistance to single agent lapatinib.

We appreciate this comprehensive and accurate summary of our study by Reviewer #2. We have addressed the specific questions raised by Reviewer #2 in the comments below.

Major points:

1. A major concern with this study is that the authors primarily used a single breast cancer cell line, MDA-MB-361, for mechanism study. This cell line harbors HER2 amplification but is intrinsically resistant to lapatinib. CCLE data indicates that this cell line harbors a large number of mutations, including BRAF(V600E) and PIK3CA(E545K). Both these mutations could explain the cell line's resistance to lapatinib. Thus, to demonstrate transaldolase to be an independent factor in lapatinib resistance and a general synthetic lethal partner of HER2 inhibitors, it would be important for the authors to demonstrate the same mechanism in additional HER2-amplified cell lines that exhibit intrinsic lapatinib resistance, preferably without known resistance mechanisms such as those mentioned above. Alternatively, would transaldolase expression in a lapatinib sensitive cell line such as BT474 confer resistance to the drug?

We thank Reviewer #2 for these insightful comments. During the revision process, we repeated most of our mechanistic studies using a second independent intrinsically resistance cell line (MDA-MB-453). First, we confirmed that TA deficiency significantly impairs cell growth and survival in the presence of lapatinib (Fig. R7a, also in revised manuscript Fig. 2c). Next, we measured cellular NADPH and NADP⁺ levels and observed that combined TA knockdown and lapatinib treatment significantly reduces NADPH levels, without significantly altering NADP⁺ (Fig. R7b, also in revised manuscript supplementary Fig. 7a-b). In addition, we detected excessive cellular ROS and increased apoptosis using the same method as previously described for MDA-MB-361 cells (Fig. R7c-d, also in revised manuscript supplementary Fig. 7d-e). In MDA-MB-

453 cells, all experiments were performed using the 1 μ M treatment condition. Together, these results further validate NADPH deficiency and redox stress as underlying mechanisms for TA's synthetic lethality with HER2 inhibition.

Fig. R7

Fig. R7 Key results repeated with a second intrinsically resistant cell line, MDA-MB-453.

Regarding BRAF and PI3K mutations, in the table below we list the mutations known to be present in these enzymes in intrinsically resistant HER2-positive cell lines.

Interestingly, PIK3CA activating mutations were identified in all intrinsically resistant HER2-positive cell lines used in previously reported studies:

Cell lines	Mutations (CCLE)
MDA-MB-361	PIK3CA(E545K); BRAF(V600E)

MDA-MB-453	PIK3CA(H1047R)
HCC1954	PIK3CA(H1047R)
UACC893	PIK3CA(H1047R)

Constitutively active PIK3CA is a known dominant factor driving resistance to HER2 inhibition and has been shown to induce resistance when expressed ectopically [7]. Notably, hexokinase activity has been previously shown to be regulated by Akt [5], so crosstalk between PI3K/Akt signaling and glycolysis is known to occur. In the current study, we did not explore whether the activities of ox-PPP and non-ox-PPP enzymes are regulated by PIK3CA, but instead focused on TA's synthetic lethality with HER2 inhibition, but this will be an interesting angle to explore in future studies. More broadly, the reviewer's point is well taken, and we agree that it will be interesting to understand how pervasive a role TA plays in intrinsic resistance, and whether this role coincides with the presence of specific genomic alterations, once sufficient numbers of intrinsically resistant cellular models lacking PIK3CA mutations become available.

Per Reviewer #2's suggestion, we also overexpressed TA in the lapatinib-sensitive cell lines BT474 and AU565. This ectopic TA expression was unable to induce anti-HER2 drug resistance (Fig. R8), perhaps because these sensitive lines already express TA protein at levels comparable to certain resistant lines (Fig. R6).

Fig. R8

Fig. R8 Ectopic expression of TA cannot induce resistance to HER2 inhibitor.

2. Mechanistically, the authors postulated that tansaldolase knockdown could perturb three pathways that synergize with lapatinib: ROS elevation, deficient lipid synthesis, and reduced nucleotide synthesis. Some of these hypotheses could be tested by chemical rescue experiments *in vitro*. Can the synthetic lethal mechanism between TA and lapatinib be rescued by treating cells with media supplemented with N-acetyl cysteine, free fatty acids such as palmitate and/or cell permeable nucleosides?

We agree with Reviewer #2 that the suggested metabolic rescue experiments would provide evidence to further validate our mechanistic studies. To test which NADPH downstream metabolites are essential for cell survival under lapatinib treatment, we added NAC (N-acetylcysteine) to scavenge cellular ROS, or used fatty acid (Sigma F7050) or nucleoside (Millipore ES-008-D) supplementation to provide cells with free fatty acids and nucleosides (Fig. R3 above, also shown in revised manuscript Fig. 5i, 5h and supplementary Fig. 7f).

The addition of NAC increased cell survival upon combinational TA and HER2 inhibition, further supporting the notion that excessive ROS is a key factor limiting the growth of

lapatinib-treated cells. We also observed a mild rescue effect by adding fatty acid supplementation to these cultures, which is consistent with a previous study showing that fatty acid synthesis is essential for HER2-positive breast cancer cell survival [3, 4]. Finally, we added nucleosides into media and did not observe enhanced cell growth, although it is possible that combining nucleosides with fatty acids and/or ROS scavengers might provide a stronger rescue. From these results, we conclude that excessive cellular oxidative stress and reduced fatty acid synthesis are consequences of NADPH shortage mediating breast cancer cell sensitivity to lapatinib.

Minor points

1. For the sgRNA screen, both DMSO and lapatinib treated cells were cultured for 2 weeks. How many population doublings were there for each arm?

During the 2-week culture, there were four population doublings for the DMSO arm and approximately two for the lapatinib treatment arm. MDA-MB-361 cells double their population every 3-4 days. This time point for sample collection was designed based on our previous experimental experiences [8]. We also collected samples at the 4-week time point, but gene depletion consistency for those samples was not sufficient for further investigation. We acknowledge that, in general, allowing for more population doublings can increase the resolution of hit identification in a screen. In this case, likely because of the strong effects of TA on lapatinib sensitivity, we could see effects after only a small number of doublings.

2. For Figure 1C, it would be more informative to plot the average of the two replicas for the DMSO treated samples vs. lapatinib treated samples. This will give a better illustration for which genes are straight lethal and which ones only dropout from lapatinib treated samples. Similarly, for Figure 1D, Figure S1D-G, they should be re-plotted in the format as Figure S3E.

We appreciate Reviewer #2's suggestion and changed the visualization of the sensitivity genes in Figure 1c and revised the description of these findings in the manuscript accordingly (Figure R9, also in revised manuscript Fig. 1c). The depletion of genes in

DMSO compared to T=0 is now plotted on the X-axis and lapatinib treatment compared to T=0 is plotted on the Y-axis. Most genes have a similar depletion ratio as they lie on the diagonal line, while IGF1R and TALDO1 were depleted more in lapatinib treatment. We revised Fig. 1d and Supplementary Fig. 1d-g to show these four conditions as well.

Fig. R9

Fig. R9 Identification of genes mediating lapatinib sensitivity.

3. Figure S2, it would be useful to indicate the IC50 values for each condition.

All IC50 values are now indicated within the figures for clarification and the method used for IC50 value calculation is explained in the revised Methods section.

4. Figure 2B showed that HER2 phosphorylation was completely inhibited by lapatinib at 1uM. Some experiments in the paper were done at either 4uM or 10uM of lapatinib. This raises the concern that whether the higher concentrations of lapatinib used could introduce off-target effect on other tyrosine kinases such as IGF1R. The author should be able to observe the same synthetic lethal effect at 1uM of lapatinib, or alternatively, with trastuzumab as an independent means to inhibit HER2. As a third alternative, HER2 can be inhibited by shRNA knockdown (CRISPR probably won't work since HER2 is amplified and cells might just get killed by Cas9-induced DNA damage).

We thank Reviewer #2 for mentioning lapatinib dosing in these experiments. Originally, we treated MDA-MB-361 cells with 4 μ M lapatinib because this concentration produced the greatest difference in cell survival. For this revision, we treated MDA-MB-361 and also a second lapatinib-resistant line (MDA-MB-453) with 1 μ M lapatinib and obtained essentially identical results for the synthetic lethality and mechanistic studies (Fig. R7 above, also in Fig. 2c, supplementary Fig. 7a-e).

5. For the K-M curves in Figure 2C and Figure S3F, were these univariate or multivariate analysis? Were the TA low and high patients stage-matched (i.e. no stage skewing in TA high patients)?

These plots were generated using kmplot.com and are univariate analysis. Unfortunately, this online database does not provide patient stage data, so we do not know whether there is stage skewing in TA high patients; however, this is an interesting question for future investigations.

6. Figure 3B and Figure S4A, in Figure 3B, M+0 labelled R5P is elevated by lapatinib; but in Figure S4A total R5P level is reduced by lapatinib. Along this same line, in Figure 4F, IMP, GMP and AMP levels are down with lapatinib or with TA knockdown. It is not clear why such differences were seen and whether these experiments measured different flux of ribose and nucleotide synthesis.

We apologize for providing an unclear explanation of the metabolic profiling data. The total metabolite level is calculated using the sum of integrated peak area of all isotopologues of each metabolite. For ^{13}C isotopologues of each metabolite, graphs were plotted to show the ^{13}C enrichment (for each sample, adding all fractions of M+n isotopes together equals 1). Specifically, for R5P, the total level was reduced by lapatinib, but the fraction of M+0 in that sample was increased compared to DMSO. We think that this isotope fraction is a more informative indicator of the proportional influx or outflux of a given metabolite.

There are many enzymatic reactions from R5P to generate IMP, GMP and AMP, which are regulated by many other factors. Therefore, total levels of IMP, GMP and AMP are not directly reflective of R5P.

7. Figure S4A, it is not clear what 6PGLU is and why it's level is strongly inhibited by either lapatinib or TA knockdown strongly depleted 6PGLU levels. Is this important?

We apologize for not providing the full name of 6PGLU (6-phosphogluconate) and have edited the Fig. S4A legend accordingly. 6PGLU level reflects the balance of production and consumption (Fig. R4). Here we hypothesize that 6PGLU depletion is caused by the differential enzyme activity of two enzymes: G6PD (glucose-6-phosphate dehydrogenase), the enzyme generating 6PGLU from G6P, and PGD (6-phosphogluconate dehydrogenase), the enzyme consuming 6PGLU to generate ribose. In this context, we don't know whether their activities are more sensitive to the concentrations of available substrate or product. Although the 6PGLU depletion caused by HER2 inhibition or TA knockdown is an interesting observation, with our current data we cannot conclude whether 6PGLU abundance is important for our analysis.

8. Figure 3D, it is not clear whether the induction of E4P and S7P by lapatinib is a known metabolic response that has been previously published? If so what is the mechanism? In addition, it is also unclear how the transaldolase-dependent induction of E4P and S7P upon lapatinib treatment is related to change in NADPH levels in Figure 4.

We apologize for the confusion. The total levels of E4P and S7P decreased with lapatinib treatment but increased with TA KD and further increased in the combined condition (Fig. R4). To our knowledge these are novel findings as these metabolites have not been previously shown to be altered by HER2 inhibition. However, induction of S7P by TA deficiency was previously shown in a TA knockout mouse model [9], which is consistent with our results.

Because of the reversible nature of the non-ox-PPP, it is complicated to directly dissect meaningful conclusions from the E4P and S7P data. Therefore, we first conclude that the non-ox-PPP flows into gluconeogenesis for NADPH generation from landscape

analysis of all metabolites (arrow in Fig. R5). Then confirmed this finding using directly measurement of NADPH level in other experiments. Specifically, for E4P and S7P, the M+1 portion increase suggests the influx from R5P towards glycolysis upon lapatinib treatment ; meanwhile, shTA reversed this increase, indicating successful blockade of the pathway, inhibiting the non-ox-PPP fueling into gluconeogenesis for ox-PPP and generation of NADPH. We labeled the isotopes of E4P and S7P in Fig. R5 (also shown in supplementary Fig. 6) to illustrate their potential sources.

9. Figure S3A and S3B – the cell lines in the figure legend are reversed? Should the left be MB-361 and the right be BT474? It would be useful to just put cell line names directly above the bar charts for Figure S3A, S3C and S3D.

We apologize for mislabeling this figure legend; these legends are now corrected, and cell line names are labeled within the figure for clarification.

10. Figure 4G and 4H, it would be useful to provide Mean +/- SEM and p-values for these experiments.

Per Reviewer #2's suggestion, in this figure (now Fig. 5g-h), we have changed the analysis to Mean +/- SEM and noted P-values on the figure and in figure legends.

11. Since PPP enzymes are induced in BT474 cells upon lapatinib treatment, does TA knockdown further sensitize this cell line towards low dose lapatinib?

We tested this possibility using three lapatinib-sensitive cell lines (BT474, AU565 and UACC812; Fig. R2 above). However, the sensitivity of these cells to lapatinib was not heightened after depleting TA. The induced expression of PPP enzymes was detected at early time point (24 hours), suggesting PPP activation as a mechanism to combat HER2 inhibition. However, activation of this pathway in sensitive cells is insufficient to rescue cell growth with predominant HER2 signaling inhibition such that TA loss-of-function cannot further sensitize these cells to lapatinib.

Reviewer #3 (Remarks to the Author):

The authors perform a CRISPR selective vulnerability screen to identify targets that will be synthetic lethal with HER2 inhibition in breast cancer.

The biological question is important. The emergence of CRISPR screening has proven to be superior over previous RNAi based method, thus the paper is timely and of interest to a wide community.

Specific comments:

1. Figure 1C - The description of this figure in the text, x and y labels and the figure labels does not match. The text refers to genes which the x and y labels refer to sgRNAs, also the legends refer to the ratio of late to early time point, in this case essential genes should have been visible on this plot. I assume that this plot represents the ratio between drug to DMSO.

We apologize for not making clear of this panel and thank Reviewer #3 for pointing this out. To better visualize the screening results, we revised this panel with a new X-axis showing the ratio of depletion for DMSO compared to T=0 and a Y-axis showing the ratio of depletion for lapatinib treatment compared to T=0. In this presentation, both IGF1R and TALDO1 stand out from all other genes (Fig. R9, also in revised manuscript Fig. 1c).

2. Not clear why the authors focused only on the five top hits, where only those significant?

We thank Reviewer #3 for this comment and have clarified our hit selection approach. To identify sensitizer genes, we calculated the fraction of sgRNAs in lapatinib (2 weeks) compared to DMSO (2 weeks). Theoretically, genes with a fraction smaller than 1.0 could be sensitizers, and a smaller number indicates a better sensitizing activity. From this analysis, 211 out of 378 genes had a fraction smaller than 1.0, 14 genes smaller than 0.9, and 5 genes smaller than 0.85, as shown in revised Fig. 1c. Among the hits smaller than 0.85, IGF1R and TALDO1 were the top hits with fractions of 0.62 and 0.77, respectively. Based on this analysis, we cloned and tested sgRNA plasmids for the five top hits.

3. Based on figure 1E it appears that TA KD affects also cells without lapatinib, would this affect its utility as a drug target? On the same line, while figure 1C nicely emphasizes the increased effect of specific genes with lapatinib compared to control, it does not provide any information on whether those genes are also essential for growth in general. A plot of the ratio between late to early time point comparing DMSO to lapatinib would help to clarify this.

We appreciate Reviewer #3's suggestion regarding plotting between late to early time points. These changes in sensitivity genes are now plotted on the X-axis in revised Fig. 1c (Fig. R9), which shows that IGF1R and TA are the two strongest candidate genes mediating sensitivity.

We understand Reviewer #3's concern that TA might be essential for normal cell proliferation, and therefore a poor therapeutic target to combine with HER2 inhibition. In response to this concern, we explored the literature and found that double allele TA knockout mice develop normally with abnormalities only in sperm motility [10], suggesting that TA is not essential for either mammalian embryonic development or somatic organ function. Therefore, it is possible that combinatorial TA and HER2 targeting might trigger stasis and/or apoptosis in HER2-positive breast cancer cells with relatively lesser effects on normal cells and tissues.

4. Figure 1F - does not match the description in the text, the actual figure is not referenced in the text

We apologize for this mistake and have corrected this in the revised manuscript.

5. The authors should show that the synthetic lethality between HER2 inhibition and TA KO/KD does not occur (or to a much-reduced extent) in cells that are not resistant to HER2 inhibition

To investigate whether synthetic lethality also occurs in sensitive cells, we cultured three lapatinib-sensitive cell lines with TA depletion in increasing concentrations of lapatinib. These experiments revealed that TA knockdown does not further sensitize

BT474, AU565, or UACC812 cells to lapatinib treatment (Fig. R2 above). In sensitive breast cancer cells, major intracellular signaling pathways, metabolism, and other essential processes are profoundly influenced by HER2 signaling. Therefore, in such cells, PPP flux may already be impaired such that further impeding this pathway via TA knockdown does not cause a further decrease in cell survival. In contrast, in intrinsically resistant cells, inhibition of HER2 signaling alters cellular metabolism, but is not sufficient to completely inhibit cell growth. Our results show that TA and PPP metabolic flux under these conditions are necessary for NADPH generation, which is vital for cell survival and proliferation.

6. It is not clear why the authors analyzed TA expression between resistance and not resistance cell lines at the mRNA level and then analyzed if lapatinib affects TA levels at the mRNA levels. It would be useful to show the former using protein levels as well.

To more thoroughly investigate TA expression as it relates to anti-HER2 drug resistance, we evaluated TA protein levels in five HER2-positive breast cancer cell lines (Fig. R6a above, also shown in revised manuscript Fig. 3a). While resistant MDA-MB-453 cells express the highest level of TA, there is no obvious correlation between basal TA expression and lapatinib sensitivity. Given the limited number of cell lines available, to better address this concern we performed immunohistochemical staining of an additional 24 clinical tumor samples from patients who received neoadjuvant trastuzumab with chemotherapy, following NCCN guidelines. Of the 44 total HER2-positive biopsies analyzed, TA expression in unresponsive (resistant) patients was significantly higher than in responsive (sensitive) patients (Fig. R6b above). This result demonstrates that TA protein expression is correlated with patient response to HER2 inhibition.

1. Xia, W., et al., *A model of acquired autoresistance to a potent ErbB2 tyrosine kinase inhibitor and a therapeutic strategy to prevent its onset in breast cancer*. Proc Natl Acad Sci U S A, 2006. **103**(20): p. 7795-800.

2. Konecny, G.E., et al., *Activity of the dual kinase inhibitor lapatinib (GW572016) against HER-2-overexpressing and trastuzumab-treated breast cancer cells*. *Cancer research*, 2006. **66**(3): p. 1630-1639.
3. Alwarawrah, Y., et al., *Fasnall, a Selective FASN Inhibitor, Shows Potent Anti-tumor Activity in the MMTV-Neu Model of HER2(+) Breast Cancer*. *Cell Chem Biol*, 2016. **23**(6): p. 678-88.
4. Vazquez-Martin, A., et al., *Pharmacological blockade of fatty acid synthase (FASN) reverses acquired autoresistance to trastuzumab (Herceptin by transcriptionally inhibiting 'HER2 super-expression' occurring in high-dose trastuzumab-conditioned SKBR3/Tzb100 breast cancer cells*. *Int J Oncol*, 2007. **31**(4): p. 769-76.
5. Majewski, N., et al., *Hexokinase-mitochondria interaction mediated by Akt is required to inhibit apoptosis in the presence or absence of Bax and Bak*. *Mol Cell*, 2004. **16**(5): p. 819-30.
6. Pusapati, R.V., et al., *mTORC1-Dependent Metabolic Reprogramming Underlies Escape from Glycolysis Addiction in Cancer Cells*. *Cancer Cell*, 2016. **29**(4): p. 548-62.
7. Berns, K., et al., *A functional genetic approach identifies the PI3K pathway as a major determinant of trastuzumab resistance in breast cancer*. *Cancer Cell*, 2007. **12**(4): p. 395-402.
8. Anderson, G.R., et al., *A Landscape of Therapeutic Cooperativity in KRAS Mutant Cancers Reveals Principles for Controlling Tumor Evolution*. *Cell Rep*, 2017. **20**(4): p. 999-1015.
9. Hanczko, R., et al., *Prevention of hepatocarcinogenesis and increased susceptibility to acetaminophen-induced liver failure in transaldolase-deficient mice by N-acetylcysteine*. *J Clin Invest*, 2009. **119**(6): p. 1546-57.
10. Perl, A., et al., *Transaldolase is essential for maintenance of the mitochondrial transmembrane potential and fertility of spermatozoa*. *Proc Natl Acad Sci U S A*, 2006. **103**(40): p. 14813-8.

REVIEWERS' COMMENTS:

Reviewer #1 (Remarks to the Author):

I have no further comments except for the following:

- While the authors assessed TA protein expression between resistant and sensitive tumor, I suggest to show individual data points and an xy graph.
- The rescue with fatty acids is barely significant $p=0.03$ and the authors claim that fatty acids may also be limiting. I suggest to repeat these experiments or soften this claim.

Reviewer #2 (Remarks to the Author):

In the revision, the authors have addressed most of the issues I raised. The new data with additional cell lines were helpful in supporting the central conclusion of the paper. The authors also showed that elevated ROS level is responsible for the combination synergy of TA and lapatinib. This provides mechanistic clarification. Figures R1 and R2 could be included as supplemental figures in the paper because they provided clarification that the combination synergy with TA knockdown only occurs in intrinsically resistant cells but not in sensitive cells or those with acquired resistance. In my opinion, the revised manuscript is now acceptable for publication.

Reviewer #3 (Remarks to the Author):

The authors have addressed my concerns

REVIEWERS' COMMENTS FOR THE REVISED MANUSCRIPT:

Reviewer #1 (Remarks to the Author):

I have no further comments except for the following:

-While the authors assessed TA protein expression between resistant and sensitive tumor, I suggest to show individual data points and an xy graph.

We changed the panel of figures into a dot plot to show individual data point.

- The rescue with fatty acids is barely significant $p=0.03$ and the authors claim that fatty acids may also be limiting. I suggest to repeat these experiments or soften this claim.

To address this question, we repeated this experiment two more times and replaced the panel with a new panel Fig. 5J with a p-value less than 0.01.

Reviewer #2 (Remarks to the Author):

In the revision, the authors have addressed most of the issues I raised. The new data with additional cell lines were helpful in supporting the central conclusion of the paper. The authors also showed that elevated ROS level is responsible for the combination synergy of TA and lapatinib. This provides mechanistic clarification. Figures R1 and R2 could be included as supplemental figures in the paper because they provided clarification that the combination synergy with TA knockdown only occurs in intrinsically resistant cells but not in sensitive cells or those with acquired resistance. In my opinion, the revised manuscript is now acceptable for publication.

We thank reviewer #2 for the suggestion. Figures R1 and R2 are now included in supplementary figures.

Reviewer #3 (Remarks to the Author):

The authors have addressed my concerns